# LNP-RNA-engineered adipose stem cells for accelerated diabetic wound healing

Yonger Xue [1,2,12], Yuebao Zhang [1,12], Yichen Zhong[1,2,12], Shi Du[1], Xucheng Hou[2], Wenqing Li[1], Haoyuan Li[2], Siyu Wang[2], Chang Wang[2], Jingyue Yan[1], Diana D. Kang[1], Binbin Deng[3], David W. McComb [3,4], Darrell J. Irvine [5,6,7,8,9], Ron Weiss [5,10,11] & Yizhou Dong [1,2] ✉

Adipose stem cells (ASCs) have attracted considerable attention as potential therapeutic agents due to their ability to promote tissue regeneration. However, their limited tissue repair capability has posed a challenge in achieving optimal therapeutic outcomes. Herein, we conceive a series of lipid nanoparticles to reprogram ASCs with durable protein secretion capacity for enhanced tissue engineering and regeneration. In vitro studies identify that the isomannide-derived lipid nanoparticles (DIM1T LNP) efficiently deliver RNAs to ASCs. Co-delivery of self-amplifying RNA (saRNA) and E3 mRNA complex (the combination of saRNA and E3 mRNA is named SEC) using DIM1T LNP modulates host immune responses against saRNAs and facilitates the durable production of proteins of interest in ASCs. The DIM1T LNP-SEC engineered ASCs (DS-ASCs) prolong expression of hepatocyte growth factor (HGF) and C-X-C motif chemokine ligand 12 (CXCL12), which show superior wound healing efficacy over their wild-type and DIM1T LNP-mRNA counterparts in the diabetic cutaneous wound model. Overall, this work suggests LNPs as an effective platform to engineer ASCs with enhanced protein generation ability, expediting the development of ASCs-based cell therapies.

Adipose stem cells (ASCs) have exhibited tremendous therapeutic potential in regenerative medicine owing to their self-renewal capability, low immunogenicity, and, most importantly, immunomodulating properties[1–7]. Compared with other multipotent somatic stem cells, ASCs are relatively convenient to be obtained and purified from adipose tissues distributed in different regions of human bodies[4,5,8].

When transplanted at damaged or diseased sites, ASCs can effectively interact with the pathological microenvironment and act through autocrine and paracrine pathways to modulate local immunity and facilitate tissue repair[9–14]. A large panel of growth factors and cytokines can be efficiently secreted by ASCs after immune licensing to fulfill various treatment criteria. Interestingly, numerous clinical studies

[1]Division of Pharmaceutics & Pharmacology, College of Pharmacy, The Ohio State University, Columbus, OH, USA. [2]Icahn Genomics Institute, Precision Immunology Institute, Department of Immunology and Immunotherapy, Department of Oncological Sciences, Tisch Cancer Institute, Friedman Brain Institute, Biomedical Engineering and Imaging Institute, Icahn School of Medicine at Mount Sinai, New York, NY, USA. [3]Center for Electron Microscopy and Analysis, The Ohio State University, Columbus, OH, USA. [4]Department of Materials Science and Engineering, The Ohio State University, Columbus, OH, USA. [5]Department of Biological Engineering, Massachusetts Institute of Technology, Cambridge, MA, USA. [6]Koch Institute for Integrative Cancer Research, Massachusetts Institute of Technology, Cambridge, MA, USA. [7]Department of Materials Science and Engineering, Massachusetts Institute of Technology, Cambridge, MA, USA. [8]Ragon Institute of Massachusetts General Hospital, Massachusetts Institute of Technology and Harvard University, Cambridge, MA, USA. [9]Howard Hughes Medical Institute, Chevy Chase, MD, USA. [10]Synthetic Biology Center, Massachusetts Institute of Technology, Cambridge, MA 02139, USA. [11]Department of Electrical Engineering and Computer Science, Massachusetts Institute of Technology, Cambridge, MA 02139, USA. [12]These authors contributed equally: Yonger Xue, Yuebao Zhang, Yichen Zhong. ✉e-mail: yizhou.dong@mssm.edu

consistently show that the long-term engraftment of ASCs isn't detectable in the recipient tissues after transplantation, meaning that the observed therapeutic effects of ASCs are performed in a 'hit-and-run' mechanism rather than retaining and replacing the local tissues[1,2,8,15–20]. Such phenomena accredit ASCs with low risks of adverse effects and ectopic tissue formation after administration. Nevertheless, it also sets a higher demand on the protein generation ability of ASCs to accomplish efficient treatments through transient intervention[21–23].

Notably, researchers have developed various engineering strategies to empower ASCs with reinforced protein generation capacity that can release therapeutic agents at diseased sites[12,22–26]. 'Hard-to-transfect' properties of ASCs, however, markedly restrict effective and safe reprogramming of ASCs using conventional techniques, such as electroporation and lipofectamine reagents. Although viral vectors can induce long-term expression of functional proteins in stem cells including ASCs, the risk of insertional mutagenesis in the ASCs may potentially lead to malignant transformation[27]. Therefore, prior ASC-based therapies mainly employ wild-type (WT) cells as therapeutic agents in clinical trials[6,18,20,21]. The inefficient clinical translation of ASC therapies may be partly ascribable to the mediocre protein-producing and immune-modulating capacity of WT cells, which results in diminished tissue regeneration efficacy. Recent studies have revealed that lipid nanoparticles (LNPs) with delicate designs can effectively deliver RNA cargos, such as messenger RNAs (mRNAs) and self-amplifying RNAs (saRNAs), to various types of primary cells and reprogram them with desired functions[28–30]. Specifically, fine-tuning the structures of ionizable lipids, such as head groups and hydrophobic tails, critically influences the RNA delivery efficiency of the formulated LNPs and their affinity for specific cell types. Unlike viral vectors, LNP carriers exhibit great biocompatibility and biodegradability[28,31,32]. Moreover, their delivery efficiency can be readily optimized by adjusting the molar ratios of individual formulation components. Additionally, with the advancements in formulation methodologies, LNP can be manufactured at large scale with high batch-to-batch reproducibility[31]. Herein, we hypothesize that LNP-assisted delivery of mRNA or saRNA encoding specific therapeutic proteins may enable ASCs to constantly and topically secrete functional proteins to regulate the local immune signaling cascade and remodel the pathological microenvironment, thereby augmenting the therapeutic ability of ASCs.

Engineered ASCs can be used for various therapeutic indications[10,12]. One area in which those ASCs can therapeutically exert their immune-modulating and protein-generating ability as well as tissue repair capability is impaired wound healing. Impaired wound healing is a pathological process where wounds fail to undergo normal healing stages and support the coordinated local immune crosstalk, which subsequently progresses to a chronic and highly inflammatory state[33–35]. Impaired wound healing is the main trigger of diabetic foot ulcers (DFU), a severe complication that causes dramatic reduction in the quality of life for diabetic patients. In the past decades, various treatment strategies including topical covering of heal-promoting dressing and administration of therapeutic proteins have been explored in clinical practice[36–38]. Regranex®, the recombinant platelet-derived growth factor beta (PDGF-BB) topical gel, is the only U.S. Food and Drug Administration (FDA)-approved standard DFU treatment to date[39,40]. However, only 50% of early-stage DFU patients benefit from this treatment, and its efficacy in pressure or venous stasis ulcers is limited. Therefore, it is of high priority to synergistically promote wound healing and remodel the local microenvironment of diabetic wounds for augmented efficacy[40]. Bone marrow-derived mesenchymal stromal cells were previously engineered to secret vascular endothelial growth factor a (VEGFA) using an electroporated-Cas9-AAV6 platform to foster diabetic wound healing[41]. Based on prior findings, we propose that the induction of sustainable expression of secretable therapeutic proteins in primary ASCs using an LNP-mRNA/saRNA platform may maximize reprogramming efficiency and the subsequent therapeutic efficacy with negligible toxicity (Fig. 1a).

Dianhydrohexitols, a series of unique cyclic ethers originally derived from hexitol sugar alcohol, serve as critical substrates in various fields of material science due to their unique properties of rigidity, chirality, and low-toxicity[42]. Inspired by their structural features, we have previously developed a class of sugar alcohol-derived ionizable lipids (DIM, DIS, and LIS) with varying head groups and hydrophobic tails for mRNA delivery to bone marrow-derived dendritic cells (BMDCs)[43]. Among them, DIM7 LNPs exhibited superior mRNA delivery efficiency in BMDCs compared with FDA-approved LNP formulations (e.g., MC3, ALC-0315, and SM102). Given that the sugar alcohol-derived ionizable lipids have showcased proficiency in delivering mRNA to primary cells, we speculate that they may serve as promising delivery vehicles for ASCs.

In this work, we investigate the potential of sugar alcohol-derived LNPs in delivering mRNAs to ASCs and conduct several rounds of tests and formulation optimizations. We then identify DIM1T LNPs as the optimal lipid formulation, which displays substantially higher delivery efficiency in ASCs than electroporation, Lipofectamine™ 3000, and three FDA-approved LNP formulations. Next, when comparing the delivery efficiency of mRNA and saRNA in ASCs, we observe that the translation of saRNA in ASCs is abruptly terminated around two days post-treatment, which could be attributed to double-stranded RNA (dsRNA) triggered cellular inflammatory responses[44–46]. Investigation of cytoplasmic level of dsRNA-sensor protein kinase R (PKR) and its downstream effector eukaryotic initiation factor 2 alpha subunit (eIF2α) confirm that the dsRNA intermediates generated by saRNA activate phosphorylation of PKR and eIF2α, resulting in termination of translation of saRNA. To overcome this challenge, we incorporate an additional mRNA encoding an immune-evasive protein E3 in the saRNA-LNP formulation, from which the target protein expression transferred by saRNA in ASCs is dramatically increased and prolonged compared with mRNA and saRNA alone treatments, respectively. In an impaired wound healing model established in db/db mice, the DIM1T-saRNA/E3 mRNA complex (SEC) LNP-engineered ASCs (DS-ASCs) significantly increase wound healing rates compared with WT ASCs after embedding on the wound sites. In summary, we develop a lipid nanoparticle strategy to engineer ASCs with durable secretion of therapeutic proteins to promote wound healing in diabetic mice. This work presents a promising stem cell engineering platform that allows convenient and efficient reprogramming of ASCs to treat diabetic ulceration, and such a strategy can be potentially further applied to treat a wide range of diseases.

## Results

### Characterizations of sugar-alcohol-derived LNPs for mRNA delivery in ASCs

Dianhydrohexitols, including isosorbide, isoidide, and isomannide, are a series of six-carbon sugar-derived heterocyclic compounds consisting of two cis-fused tetrahydrofuran rings with two secondary hydroxyl groups situated at carbon 2 and 5[42]. The different configurations of the two hydroxyl groups, designated as exo or endo, generate three isomers with distinct reaction activity and steric hindrance. Such structure diversity empowers dianhydrohexitols as versatile building blocks for the development of new chemicals in the medical and material sciences. Inspired by the unique properties of dianhydrohexitols, three sets of ionizable lipids (DIM, DIS, and LIS) were prepared with isomannide, isosorbide, and L-sorbitol as the starting materials, respectively (Supplementary Fig. 1). The synthetic route of DIM lipids was shown in Fig. 1b as a general synthetic approach to all sets of lipids. Commercially available isosorbide was treated with acrylonitrile and sodium hydroxide to give compound 3 via a double Michael addition reactions. Then, the free core amine 4 was obtained

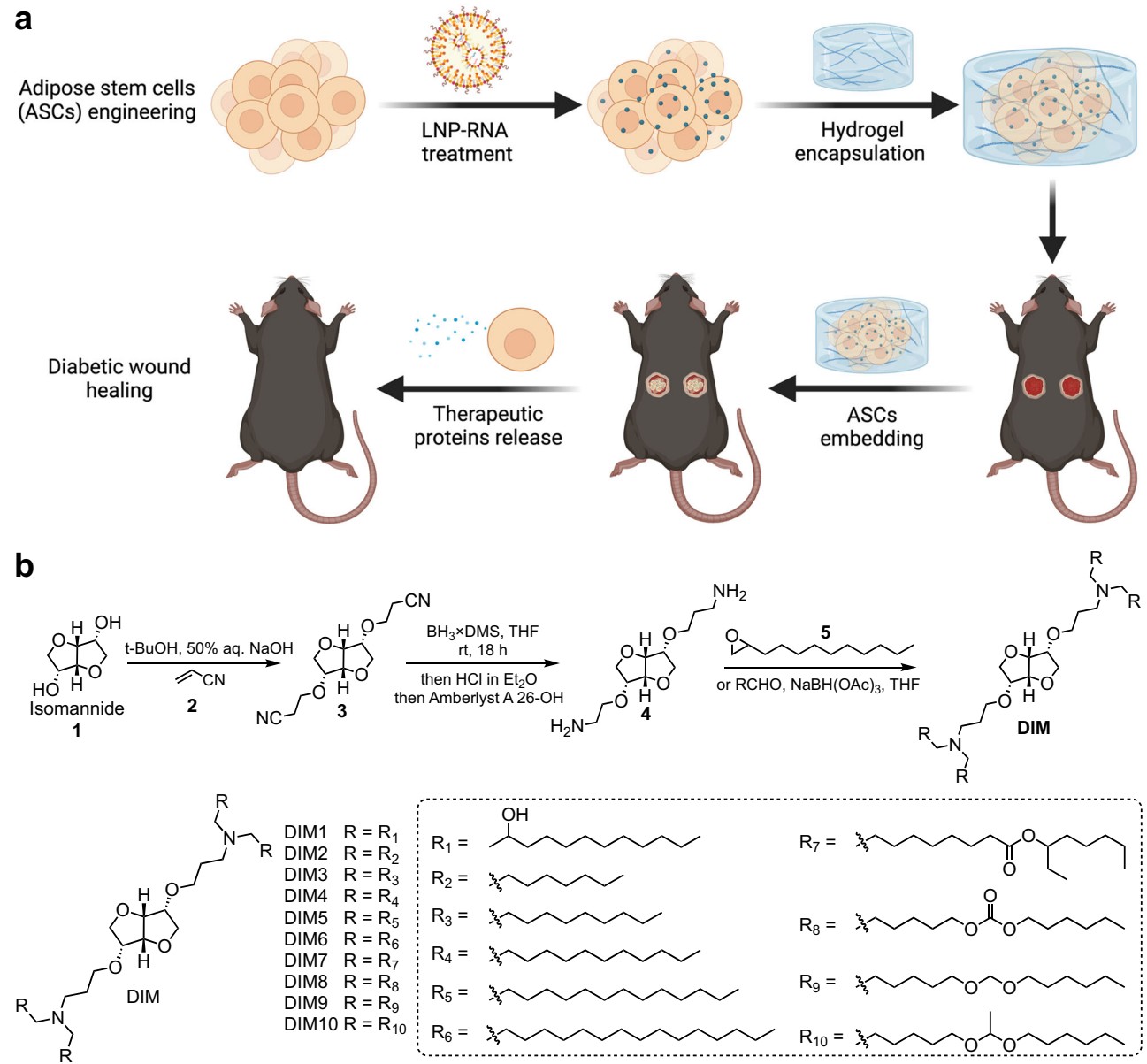

**Fig. 1 | LNP-RNA-engineered adipose stem cell therapy to treat acute diabetic wounds. a** Illustration of LNP-engineered ASCs with enhanced protein-generating ability for diabetic wound healing. This illustration was created with BioRender.com and permitted for publication. **b** A representative synthetic route for sugar alcohol-derived ionizable lipids and structures of isomannide-derived ionizable lipids (DIM lipids).

via the reduction of the nitrile groups with borane followed by treatment with hydrochloride and anionic ion exchange resin[47]. As the hydrophobic tails of ionizable lipids can greatly affect LNP formulations and their interaction with bio-membranes such as cell membranes and endosome membranes, we installed five types of hydrophobic tails to the core amine 4 to equip DIM lipids via epoxide ring-opening reaction with alkyl epoxide or reductive amination reaction with aldehydes. The resulting lipids contain hydrophobic domains incorporating five different functional groups, such as hydroxyl (DIM1), hydrocarbon (DIM2 to DIM6), ester (DIM7), carbonate ester (DIM8), and acetal (DIM9 and DIM10) (Fig. 1b; Supplementary Fig. 1). The synthesis of LIS lipids started with the treatment of L-sorbitol with sodium ethoxide and dimethyl carbonate, resulting in the enantiomer of isosorbide[48]. With the commercially available isosorbide (derived from D-sorbitol) and the enantiomer of isosorbide as the starting materials, the DIS and LIS lipids were synthesized following similar synthetic routes, respectively. [1]H nuclear magnetic resonance

and mass spectrometry were used to confirm the chemical structures of these ionizable lipids[43].

As reported in our prior studies[43,49], we formulated these dianhydrohexitols-derived ionizable lipids with firefly-luciferase (FLuc) mRNAs (FLuc-LNPs) and evaluated their physicochemical properties (Supplementary Fig. 2a–c). We then examined their mRNA delivery in ASCs based on luminescence intensity and profiled the structure-activity relationship. Generally, isomannide-based lipids (DIM) showed more efficient mRNA delivery than DIS and LIS lipids, the amine cores of which are a pair of enantiomers, indicating that the chirality of the amine cores can influence the delivery efficiency of the formulated LNPs in primary ASCs (Fig. 2a). Moreover, the lipids with hydroxyl tails, including DIM1, DIS1, and LIS1, yielded higher mRNA delivery capacity than those with other hydrophobic tails. In particular, the LNPs with hydrocarbon and carbonate ester groups in tails showed negligible mRNA delivery efficiency in ASCs. Among all these LNPs, DIM1 LNPs exhibited the highest capacity to deliver mRNA, which was over

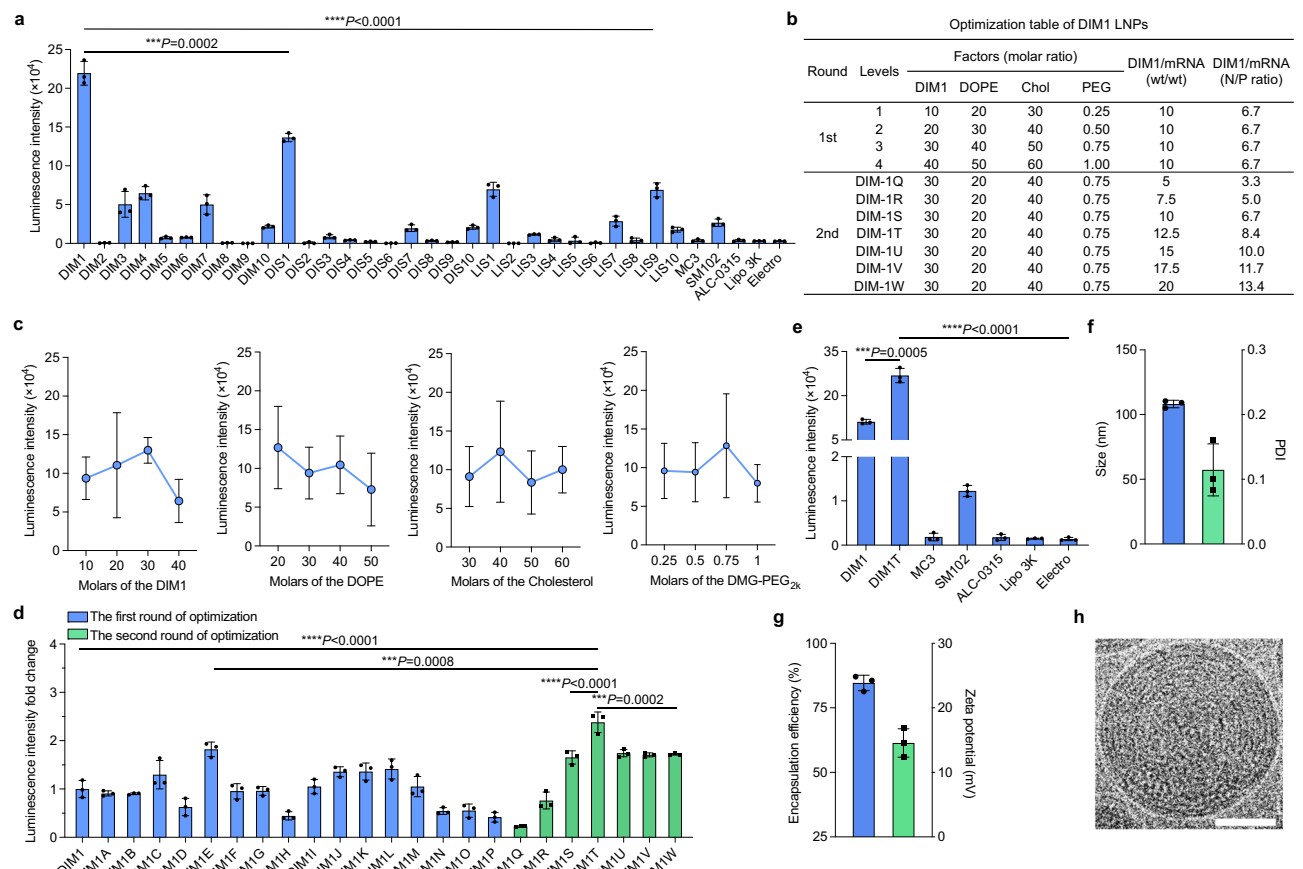

**Fig. 2 | Screening, optimization, and characterization of sugar alcohol-derived lipid nanoparticles. a** FLuc-mRNA delivery efficiency in primary murine ASCs represented as luminescence intensity. Data are from $n = 3$ biologically independent samples. **b** Table for the two rounds of DIM1/FLuc-mRNA LNP optimization. Chol cholesterol, PEG DMG-PEG$_{2k}$. **c** Orthogonal assay to determine impact trends of each lipid component in DIM1 formulation at four levels. **d** Luminescence intensity fold changes of the two rounds of optimization. **e** Luminescence intensity of DIM1T LNPs and other control groups. **f** Hydrodynamic diameter (blue) and PDI (green) of DIM1T LNPs. **g** Encapsulation efficiency (blue) and zeta potential (green) of DIM1T LNPs. **h** Cryo-TEM image of DIM1T LNPs (scale bar = 50 nm). Data in **h** are representative images from $n = 3$ independent experiments. Data in **a** and **c–g** are from $n = 3$ biologically independent samples and are presented as mean ± standard deviation (s.d.). One-way ANOVA followed by Dunnett's multiple comparison test is used to determine the statistical significance and $P$ values. ***$P < 0.001$, ****$P < 0.0001$.

70-fold more effective than Lipofectamine™ 3000 and electroporation at the same mRNA concentration (Fig. 2a). To optimize the formulation of DIM1 LNPs, we designed an L16 (4)[4] orthogonal table for the determination of optimal molar ratios of each lipid (Supplementary Fig. 3a). Then, we varied the mass ratios of DIM1 lipid to mRNA in the orthogonal-predicted formulation to further increase the mRNA delivery efficiency of DIM1 LNPs (Fig. 2b, c). The optimal formulation DIM1T resulted in 1.5-fold and 2.7-fold higher luminescence intensity than the top orthogonal formulation DIM1E and the original formulation DIM1, respectively (Fig. 2d). Moreover, the delivery efficiency of DIM1T LNPs in ASCs was over 140-fold greater than those of ALC-0315 and MC3 LNPs, and 22-fold higher than that of SM102 LNPs (Fig. 2e). In addition, DIM1T LNPs encapsulating GFP mRNA yielded over 90% GFP positive ASCs, with the mean fluorescence intensity (MFI) notably superior to those FDA-approved lipid formulations (Supplementary Figs. 4a–c and 5a). The hydrodynamic diameter of DIM1T LNPs was around 110 nm with a polydispersity index (PDI) < 0.15 (Fig. 2f). Over 80% mRNA was encapsulated in DIM1T LNPs, and the resultant particles displayed a slightly positive charge and spherical morphology (Fig. 2g, h; Supplementary Fig. 3b). Additionally, DIM1T LNPs had an apparent pK$_a$ of 6.56 as analyzed by the 6-(p-toluidino)−2-naphthalenesulfonic acid (TNS) assay (Supplementary Fig. 3c)[50]. Accordingly, the DIM1T LNPs were chosen for ex vivo ASCs engineering in the following studies.

## Delivery of saRNA and E3 mRNA complex (SEC) using DIM1T LNPs facilitates prolonged protein production in ASCs

Sustained protein production by ASCs is crucial for augmenting their therapeutic potency. In contrast to traditional IVT mRNAs, self-amplifying RNAs (saRNAs), which encode both genes of interest and viral replicase, can self-replicate within cells and continuously generate equivalent or superior expression of desired proteins at a lower dose[51]. Therefore, we speculate that the delivery of saRNA encoding specific therapeutic proteins to ASCs may facilitate topical protein secretion for a relatively long period of time compared with mRNA. To study the delivery feasibility of saRNA in ASCs, we quantified the luminescence intensity in the ASCs treated with DIM1T LNPs encapsulating FLuc-saRNA. Interestingly, the expression of FLuc-saRNA in ASCs plunged to a negligible level 48 h post-treatment (Fig. 3a). We reproduced this treatment in 293T cells, a human embryonic kidney cell line, and did not observe a similar sharp decrease in expression (Supplementary Fig. 6a). Such different expression dynamics indicated that certain inflammatory pathways in ASCs were triggered to shut down the translation of saRNA or even directly eliminate them since saRNA can generate double-stranded RNA (dsRNA) intermediates during replicative translation, which are natural ligands of cytoplasmic RNA sensors[46]. As previously reported, dsRNA intermediates may activate PKR and subsequently phosphorylate eIF2α, thereby blocking cap-dependent translation[45]. To explore the mechanism of saRNA

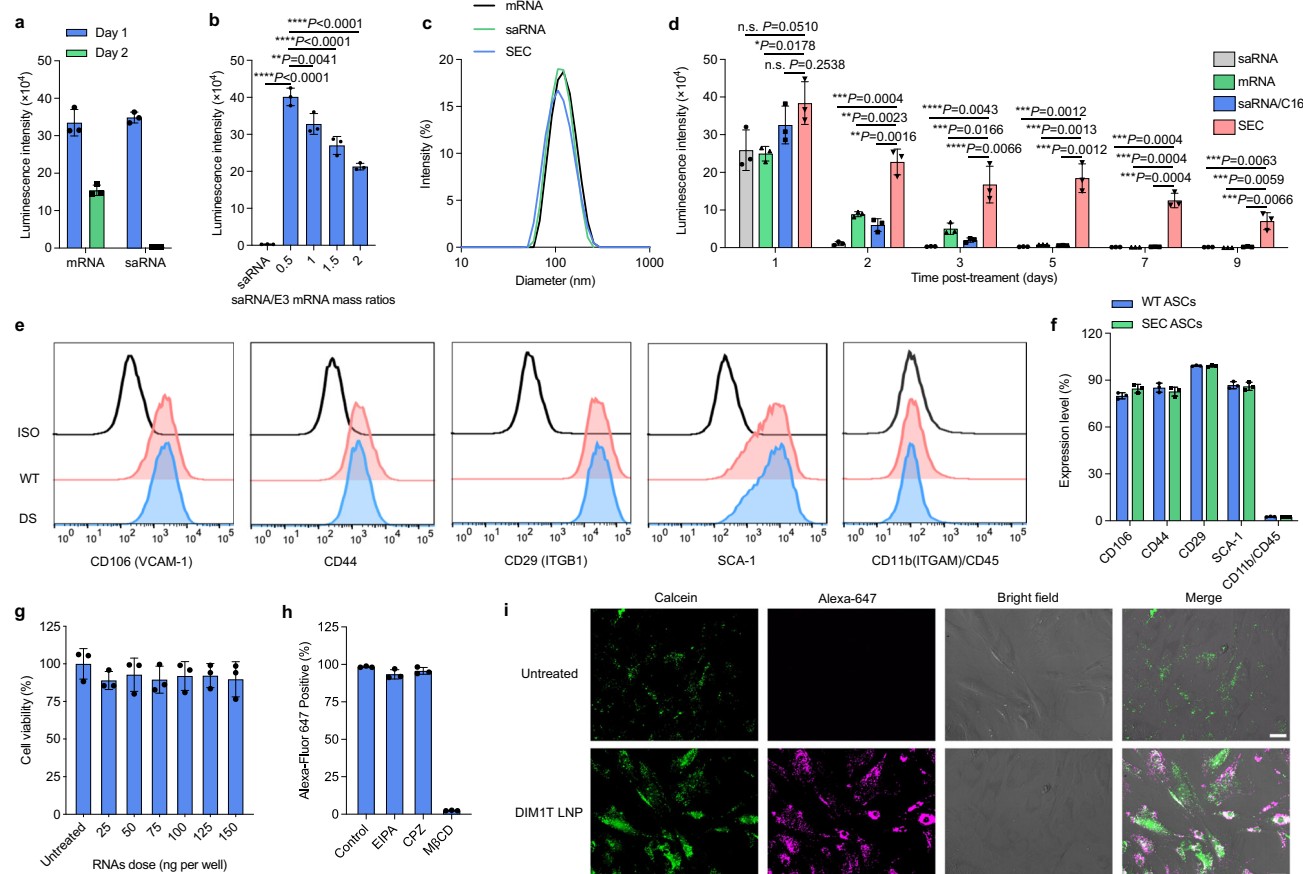

**Fig. 3 | Durable protein expression in ASCs. a** Kinetics of luminescence intensity DIM1T firefly-luciferase saRNA or mRNA LNPs treated ASCs after 2 days. **b** Luminescence intensity of DIM1T LNPs encapsulating FLuc-saRNA and E3 mRNA complex (SEC) at various saRNA/E3 mass ratios in ASCs after 48 h post-treatment. The total RNA dose is 50 ng/10000 cells. **c** Size distribution of DIM1T LNPs encapsulating FLuc mRNA, FLuc saRNA or FLuc SEC. **d** Kinetics of different DIM1T-RNAs LNPs in ASCs for 9 days. The statistical significance is analyzed using the two-tailed Student's *T* test. **e** Surface expression levels of CD106, CD44, CD29, SCA-1, CD11b and CD45 on DS-ASCs. WT ASCs and isotype staining serve as controls. CD106 vascular cell adhesion molecule 1(VCAM-1), CD29 Integrin beta 1 (ITGB1),

Sca-1 Stem Cell Antigen-1, CD11 Integrin alpha M (ITGAM). **f** Quantification of expression of each surface marker from **e**. **g** Cell viability of SEC-DIM1T LNPs in ASCs at various total RNA doses. **h** Uptake efficiency of DIM1T LNPs loaded with Alexa-Fluor 647 labeled RNAs in ASCs treated by various endocytic inhibitors. **i** CLSM images of ASCs stained with calcein alone or with DIM1T-LNPs (scale bar = 50 μm). Data in **i** are representative images from *n* = 3 independent experiments. Data in **a**, **b**, and **d**–**h** are from *n* = 3 biologically independent samples and are presented as mean ± s.d. One-way ANOVA followed by Dunnett's multiple comparison test is used to analyze the statistical significance. n.s. not significant, $P > 0.05$, *$P < 0.05$, **$P < 0.01$, ***$P < 0.001$, ****$P < 0.0001$.

translational shutdown in ASCs, we evaluated the cytosolic level of PKR, eIF2α, and their phosphorylated derivatives in the saRNA-treated ASCs. Delivery of FLuc-saRNA to ASCs sharply increased the intracellular expression of the phosphorylated PKR and eIF2α after 48 h compared with the WT ASCs and mRNA-treated ASCs, indicating that the dsRNA intermediates triggered the PKR/eIF2α-mediated translation inhibition in ASCs (Supplementary Figs. 7a–e and 8). To evade cellular immunity against saRNA translation, specialized immune evasion proteins developed by viruses can be implemented to either avoid activation of PKR or prevent eIF2α phosphorylation. It has been demonstrated that co-delivery of saRNA and mRNA encoding vaccinia virus (VACV) protein E3 could substantially increase and prolong the translation of saRNA in human foreskin fibroblasts (HFF)[44]. To evaluate whether E3 proteins can expedite the durable expression of genes transferred by saRNA in ASCs, we encapsulated FLuc-saRNA and E3-mRNA in DIM1T LNPs at different saRNA/mRNA mass ratios and treated ASCs with these LNPs at the same total RNA doses. The results showed that co-delivery of E3-mRNA significantly augmented luciferase expression by saRNA and the expression could remain at least 48 h post-treatment (Fig. 3b). Moreover, the DIM1T LNPs encapsulating FLuc-saRNA/E3 mRNA complex at a mass ratio of 0.5 induced the highest luminescence intensity among all the ratios tested (Fig. 3b). Delivery of

FLuc saRNA/E3 mRNA complex (abbreviated as SEC) using DIM1T LNPs significantly decreased the expression level of phosphorylated PKR and eIF2α in ASCs compared with that of FLuc saRNA-treated ones (Supplementary Figs. 7a–e and 8). Collectively, the saRNA/mRNA mass ratio of 0.5 was applied in the following experiments.

To investigate the expression duration of FLuc-SEC in ASCs, we treated ASCs with FLuc-mRNA, FLuc-saRNA, and FLuc-SEC at the same total RNA doses, respectively. The DIM1T LNPs displayed similar particle sizes, encapsulation efficiency and zeta potential when formulating different RNA cargos (Fig. 3c, Supplementary Fig. 6b, c). Compound C16, a PKR inhibitor, was included as a positive control group. Significantly, co-delivery of FLuc-SEC induced a 1.48-fold, 1.54-fold, and 1.17-fold higher luminescence signal than FLuc-saRNA, FLuc-mRNA, and FLuc-saRNA/C16 group on day 1, respectively (Fig. 3d). Such leads were substantially expanded on day 2 as the translation of saRNA was blocked by ASCs without inhibiting PKR activation. Compared with the C16 compound, the delivery of E3-mRNA was able to make the inhibition more durable as only the FLuc-SEC group could still detect FLuc expression on day 5, which then persisted for another 4 days. Additionally, intracellular levels of E3 proteins in ASCs remained detectable on day 9 (Supplementary Fig. 9a). Despite the low levels, these levels were adequate to retain the expression of proteins

of interest after 9 days. To evaluate whether the delivery of FLuc-SEC using DIM1T LNPs would alter the ex vivo characteristics of ASCs, we used flow cytometry analysis to compare surface marker expression between non-engineered ASCs and FLuc-SEC mRNA reprogrammed ones. Notably, we found both types of ASCs express similar levels of CD106, CD44, CD29, and SCA-1 in the absence of CD11b and CD45, indicating the treatment of FLuc-SEC mRNA delivered by DIM1T LNPs could induce and facilitate sustained protein production in ASCs without disrupting their phenotypes (Fig. 3e, f; Supplementary Fig. 5b). Moreover, both WT ASCs and their SEC-engineered counterparts retained the differentiation capacity as evidenced by the detection of FABP4+ and osteopontin+ cells after incubation in standard differentiation culture, respectively (Supplementary Fig. 9b). The DIM1T LNPs encapsulating FLuc-SEC caused minimal cytotoxicity to ASCs at various RNA doses, confirming the excellent biocompatibility of DIM1T lipids (Fig. 3g). To investigate cellular uptake pathways of DIM1T LNPs in ASCs, we incorporated an RNA labeled with Alexa-Fluor 647 in the formulation. Only when pre-incubating the cells with methyl-beta-cyclodextrin (MβCD), a caveolae-mediated endocytosis inhibitor, did we observe a dramatic decrease (97%) in uptake efficiency, suggesting that the cellular uptake of these DIM1T LNPs in ASCs is predominantly mediated by caveolae-mediated endocytosis (Fig. 3h). By treating ASCs with both DIM1T LNPs and calcein, a membrane-impermeable fluorescent dye, we observed the green fluorescence dots diffused in the cytoplasm of DIM1T LNPs-engineered ASCs, suggesting the endosome membranes were disrupted and DIM1T LNPs were released into the cytosol (Fig. 3i).

## HGF DIM1T-SEC engineered ASCs (DS-ASCs) accelerate diabetic wound healing

To explore the therapeutic potential of ASCs engineered by DIM1T-SEC LNPs in diabetic cutaneous wounds, we constructed saRNAs encoding hepatocyte growth factor (HGF) to generate therapeutic protein-hypersecreting ASCs. Previous studies reported that HGF can regulate cellular migration, proliferation, and morphogenesis in many types of cells at wound sites, thereby promoting epithelial repair and neovascularization during wound healing[52,53]. The short half-life of HGF (<3–5 min) in plasma, however, may require repeated administrations to maintain therapeutic windows in clinical applications. Therefore, the sustained production of HGF at the wound sites by ASCs may establish a clinically feasible treatment strategy. To confirm the construction of HGF-encoded saRNAs, the HGF secretion level of DS-ASCs in the conditioned medium was quantified using ELISA assay. The DIM1T-SEC-engineered ASCs secreted a significantly higher level of mature HGF proteins than the mRNA- or saRNA- treated groups from Day 1, the level of which was still detectable 9 days post-treatment (Fig. 4a). For saRNA alone groups, the secretion level of proteins was negligible on Day 2, corresponding to a similar dynamic expression pattern in the FLuc-saRNA assay. This result suggested that the translation of HGF proteins by saRNA was completely hampered in ASCs and the incorporation of E3 mRNA played an essential role in maintaining protein expression.

To evaluate the therapeutic functions of DS-ASCs, we established an excisional wound model in db/db mice, which mimics the delayed wound healing process observed in DFU patients[41,54,55]. Instead of subcutaneously administrating ASCs in the wound sites, we embedded ASCs with an in situ crosslinked HyStem®-HP hydrogel above the wound bed for sustained cell retention and HGF protein generation[41,56]. This hydrogel system is composed of crosslinked hyaluronan, heparin, and denatured collagen, and has been extensively utilized in cell culture and delivery[41,57]. The ASCs were treated with DIM1T-HGF mRNA (DM) or DIM1T-HGF SEC for 12 h before being harvested and encapsulated in the hydrogels. Shortly after wound formation, $3 \times 10^5$ DIM1T-engineered ASCs were embedded above the wound bed (n = 10 wounds per group). The wounds were imaged in

3-day intervals until closure (Fig. 4b). Starting from Day 6, we observed pronounced differences in wound healing rates between different groups. Compared with the vehicle group (hydrogel only), all ASC groups exhibited accelerated wound size reduction. Among them, the HGF DS-ASCs yielded the most efficient wound-healing kinetics as the average wound size of this group on Day 18 was 0.00 ± 0.00% (Fig. 4c; Supplementary Fig. 10a). Such rates in other groups were 25.60% ± 6.28% (Vehicles), 18.87% ± 4.70% (WT ASCs), and 7.03% ± 6.29% (HGF DM-ASCs). The area-under-curve (AUC) of each wound was graphically computed and normalized as average values against the vehicle groups. The normalized AUCs of wounds in the DS-ASCs group were significantly lower than those of wounds treated by WT ASCs and HGF DM-ASCs, respectively (Fig. 4d). Moreover, the wounds treated with HGF DS-ASCs showed complete closure starting from Day 15 and were all closed after Day 18 post-wounding, the time-to-closure function of which was significantly more efficient than other groups (Fig. 4e). The wounds treated by HGF DS-ASCs demonstrated more efficient re-epithelialization and displayed well-formed hyperproliferative epidermis compared with other groups, which is an encouraging indication of advanced healing (Fig. 4f, g). Furthermore, enhanced vascularization was observed in the wounds treated with HGF DS-ASCs as evidenced by the higher density of CD31+ vessels than other groups (Fig. 4h, i). Normally, the conversion of fibroblasts to myofibroblasts was limited in diabetic wounds, resulting in deficient wound contraction capability. However, the population of αSMA+ myofibroblasts in the wounds was substantially increased in the HGF DS-ASCs group, demonstrating a more active and efficient healing process (Fig. 4j, k). These data suggested that the delivery of HGF saRNA/E3 mRNA complex using DIM1T LNPs could elevate and elongate the secretion of HGF proteins by ASCs, thereby enhancing their therapeutic efficacy in healing diabetic wounds.

## CXCL12 DS-ASCs reprogram local immune microenvironment in diabetic wounds

In addition to growth factors, chemokines have been explored to promote diabetic wound healing[58,59]. For example, CXCL12 (also known as stromal cell-derived factor 1α) is an immune-regulating chemokine that can bind to C-X-C chemokine receptor type 4 (CXCR4) expressed on immune cells and keratinocytes at wound sites, thereby mitigating local inflammation, enhancing angiogenesis, and promoting cell proliferation[60,61]. Similar to HGF proteins, CXCL12 has a short half-life in plasma, demanding repetitive administration to reach therapeutically efficacious levels. To compare the therapeutic efficacy between HGF and CXCL12 in healing diabetic wounds, we constructed saRNA encoding CXCL12 for SEC-DIM1T-mediated ASC engineering and reproduced the excisional wound healing experiments in db/db mice using the same procedure described before. In an ex vivo protein secretion assay, the CXCL12 DS-ASCs were able to constantly generate CXCL12 in culture medium, the level of which remained detectable for 9 days post-treatment (Fig. 5a). After being embedded on the wound sites, the CXCL12 DS-ASCs reduced wound size at significantly accelerated healing kinetics (0.00% ± 0.00% in relatively wound size on Day 15) and showed substantially smaller AUCs of wounds compared with HGF-generating groups (10.04% ± 9.14% in relatively wound size on Day 15) (Fig. 5b–d; Supplementary Fig. 10b). Moreover, all the wounds treated with the CXCL12 DS-ASCs were entirely closed on Day 15 while, simultaneously, only 50% of the wounds in the HGF DS-ASCs group demonstrated complete closure (Fig. 5e). Moreover, the wounds treated by the CXCL12 DS-ASCs displayed a significantly thicker layer of epidermis at wound closure sites and a substantially higher density of CD31+ blood vessels and αSMA+ myofibroblast (Fig. 5f–k). Notably, immunofluorescence staining for proinflammatory cytokine, interleukin-6 (IL-6), and anti-inflammatory cytokine, interleukin-10 (IL-10), indicated that the CXCL12 SEC-engineered ASCs significantly

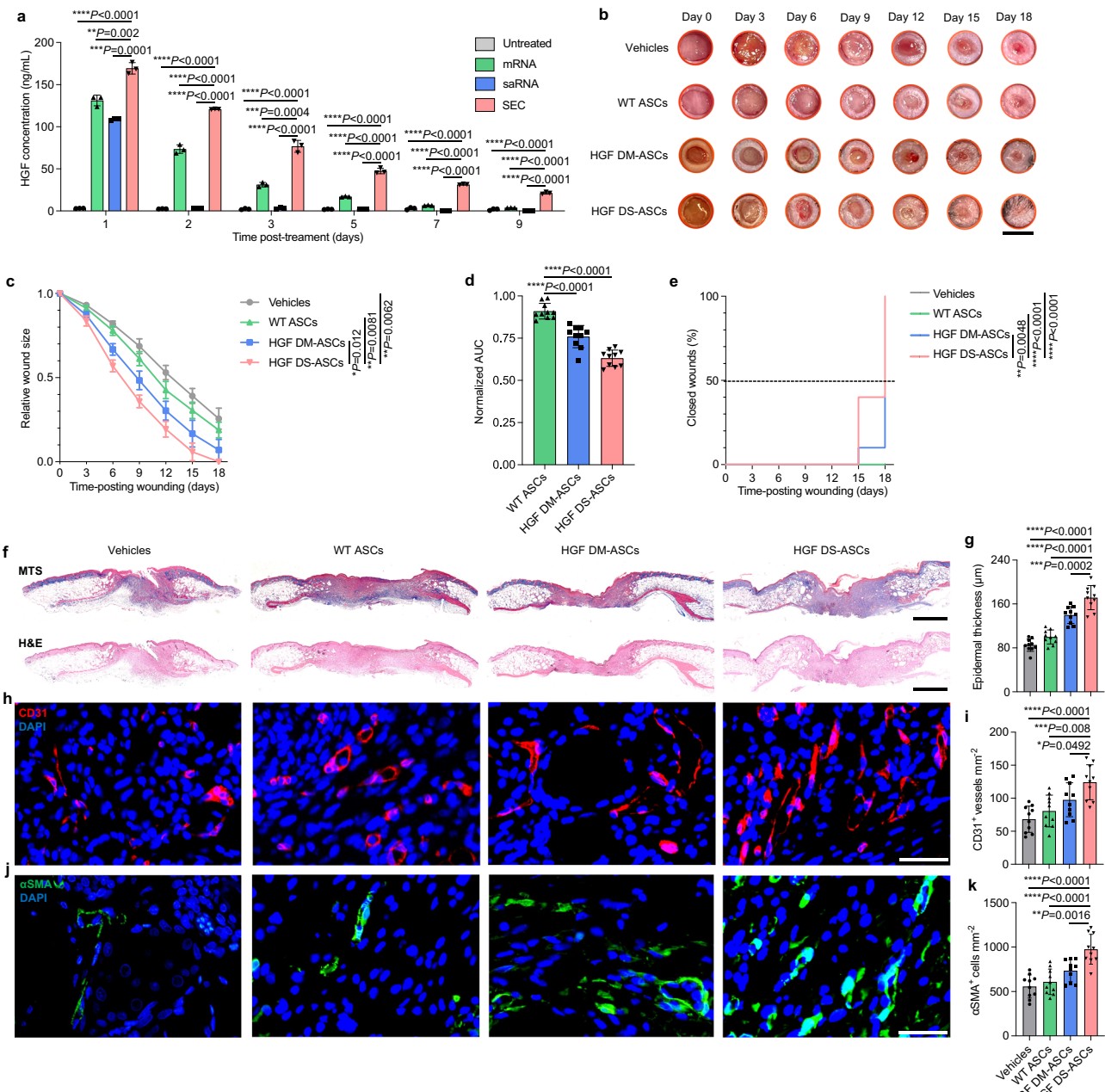

**Fig. 4 | DIM1T-HGF SEC LNPs-engineered ASCs accelerated wound healing in diabetic mice. a** Expression kinetics of HGF SEC delivered by DIM1T LNPs in ASCs. The conditioned medium was collected on Days 1, 2, 3, 5, 7, and 9. The protein level in the medium was analyzed using an ELISA kit. The statistical significance is analyzed using the two-tailed Student's T-test. Data in **a** are from $n = 3$ biologically independent samples. **b** Representative digital image of the skin wounds of each group. **c** Relative wound size of vehicle controls, WT ASCs, HGF DM-ASCs, and HGF DS-ASCs. $n = 10$ for all groups. **d** Mean AUC of individual wounds of each group. $n = 10$ for all groups. **e** The complete wound closure time in vehicle controls, WT ASCs, HGF DM-ASCs, and HGF DS-ASCs. The significant differences in time to closure between groups are analyzed using the Log-rank test. $**P < 0.01$, $****P < 0.0001$. **f** Representative Masson's trichrome staining (MTS) of wounds on D18 for each group. **h, j** Representative CD31+ and αSMA+ immunofluorescence (IF) images of wounds on D18 for each group. **g** Quantification of the epidermal thickness of the wounds from **f**. **i** Quantification of CD31+ vessels per unit area on D18 from **h**. **k** Quantification of αSMA+ cells per unit area on D18 from **j**. Data in **g, i**, and **k** are from $n = 10$ biologically independent samples. All data are presented as mean ± s.d. Statistical significance and $P$ values are analyzed by one-way ANOVA followed by Dunnett's multiple comparison test. $*P < 0.05$, $**P < 0.01$, $***P < 0.001$, $****P < 0.0001$. Scale bars, 7 mm (**b**); 1 mm (**f**); 50 μm (**h** and **j**).

mitigated the dysregulated local inflammation at the wound sites compared with HGF DS-ASCs and WT controls (Fig. 5l–o). The sustained generation of CXCL12 chemokines at wound sites by the CXCL12 DS-ASCs facilitated the effective shift from the early inflammation stage to the anti-inflammatory stage, resulting in enhanced efficacy for wound healing.

To investigate the potential synergistic effects of CXCL12 and HGF in wound healing, we conducted the same excisional wound healing experiments in db/db mice and embedded the engineered DS-ACSs generating both CXCL12 and HGF on the wound sites. In comparison with vehicle controls and FLuc DS-ASCs group, the wounds treated with the CXCL12/HGF DS-ASCs or CXCL12 DS-ASCs demonstrated pronounced acceleration in the healing process, which was highlighted by the decrease in wound size and smaller wound healing AUC (Supplementary Figs. 11 and 12a, b). Particularly, the average relative wound size of these groups on Day 15 was 28.87% ± 6.67% (Vehicles),

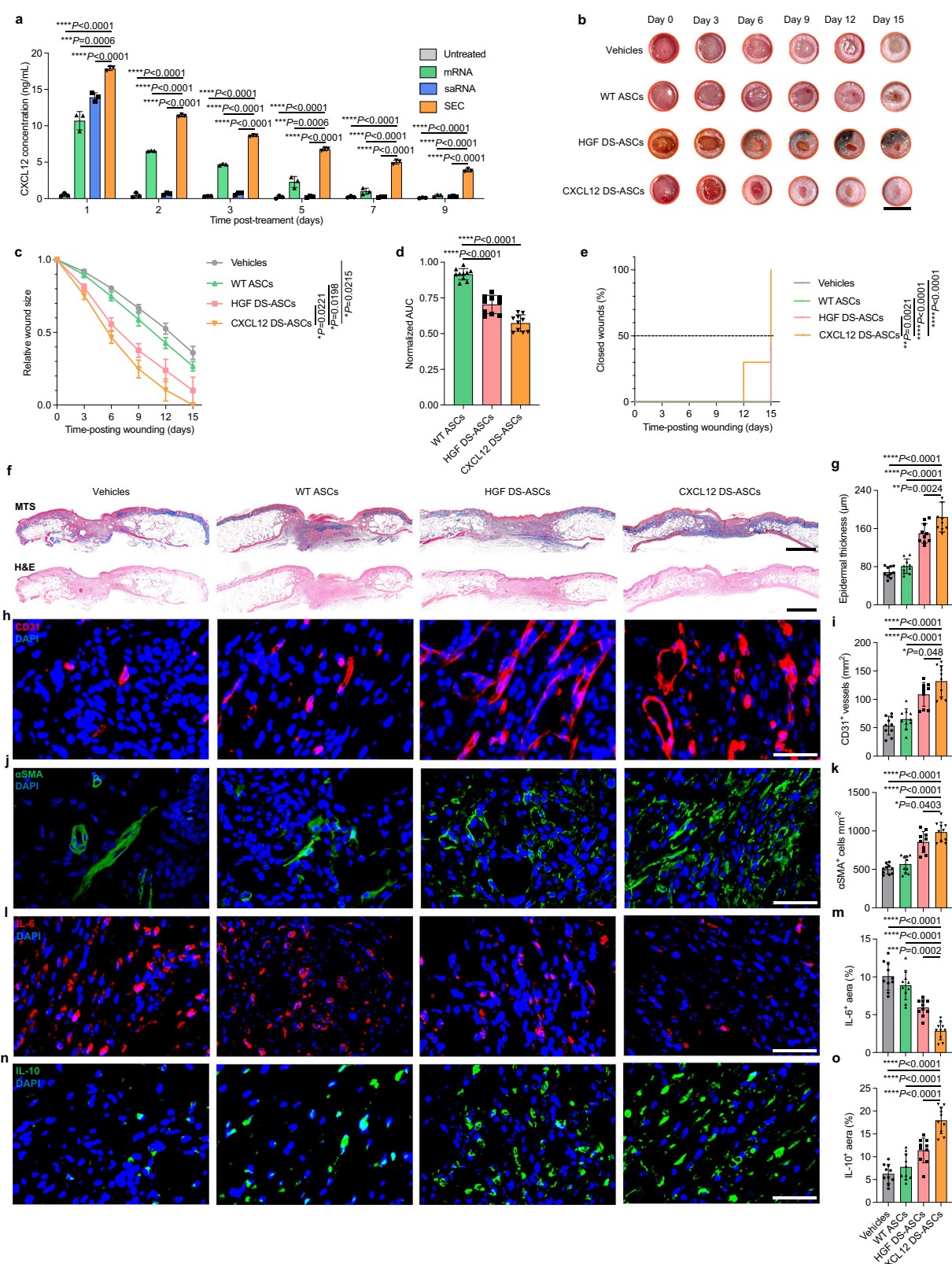

20.00% ± 7.21% (FLuc DS-ASCs), 0.00% ± 0.00% (CXCL12 DS-ASCs) and 1.3% ± 1.9% (CXCL12/HGF DS-ASCs). Remarkably, 80% of the wounds in CXCL12/HGF SC-ACSs group and a full 100% in CXCL12 DS-ASCs group exhibited complete closure by Day 15 (Supplementary Fig. 12c). Moreover, both CXCL12 DS-ASCs and CXCL12/HGF DS-ASCs substantially increased the thickness of the hyperproliferative epidermis and density of CD31⁺ blood vessels and αSMA⁺ myofibroblast relative to vehicles and FLuc-DS ASCs (Supplementary Fig. 12d–i). Both groups also showed increased accumulation of IL10 as well as a decreased level of IL6 in the wound microenvironment (Supplementary Fig. 12j–m). However, no statistically significant differences were observed in the aforementioned parameters between these two groups, suggesting the absence of synergistic therapeutic efficacy when combining CXCL12 and HGF in expediting wound healing.

**Fig. 5 | CXCL12-generating ASCs outperformed HGF counterparts in healing acute diabetic wounds. a** Expression kinetics of CXCL12 SEC delivered by DIM1T LNPs in ASCs. The statistical significance is analyzed using the two-tailed Student's T-test. Data in **a** are from $n = 3$ biologically independent samples. **b** Representative digital image of the wounds of each group. **c** Relative wound size of vehicle controls, WT ASCs, HGF DS-ASCs, and CXCL12 DS-ASCs. $n = 10$ for all groups. **d** Mean AUC of individual wounds of each group. $n = 10$ wounds per group. **e** The complete wound closure time in vehicle controls, WT ASCs, HGF DS-ASCs, and CXCL12 DS-ASCs. The significant differences in time to closure between groups are analyzed using the Log-rank test. **$P < 0.01$. ****$P < 0.0001$. **f** Representative MTS and H&E images of wounds on D15 for each group. **g** Quantification of the epidermal thickness of the wounds on D15. **h, j, l, n** Representative CD31[+], αSMA[+], IL-6 and IL-10 IF images of wounds on D15 for each group. **i, k, m, o** Quantification of the CD31[+] cells, the αSMA[+] cells, the IL-6, and the IL-10. Data in **g, i, k, m,** and **o** are from $n = 10$ biologically independent samples. All data are presented as mean ± s.d. Statistical significance and $P$ values are analyzed by one-way ANOVA followed by Dunnett's multiple comparison test. *$P < 0.05$, **$P < 0.01$, ***$P < 0.001$, ****$P < 0.0001$. Scale bars, 7 mm (**b**); 1 mm (**f**); 50 μm (**h, j, l,** and **n**).

Furthermore, to assess the potential effects of gender differences on diabetic wound healing rate, we conducted parallel investigations employing identical experimental protocols across treatment groups: vehicle controls, FLuc DS-ASCs, CXCL12 DS-ASCs, and CXCL12/HGF DS-ASCs. Each of these groups was balanced with an equal representation of male and female db/db mice (four each). Throughout the duration of the study, the relative wound size, measured on average or individually, did not reveal any significant variations in the wound healing kinetics attributable to sex differences (Supplementary Fig. 13a–d).

## Discussion

The development of adipose stem cells (ASCs) as potential therapeutic agents has gained significant attention due to their immunomodulatory and tissue regenerative properties. However, their limited protein-generating ability has posed challenges in achieving optimal therapeutic outcomes. To empower ASCs with an enhanced secreting capability of specific proteins, we developed a series of dianhydrohexitols-derived ionizable lipid formulations as the RNA-mediated engineering platform to reprogram ASCs and analyzed the structure-activity relationship of the formulated LNPs in terms of mRNA delivery efficiency. We found that the chirality of the amine cores in the head group as well as the chemical structure and length of the hydrophobic tails could considerably affect the mRNA delivery efficiency of the lipid derivatives in primary ASCs. From a series of cell studies, we identified DIM1T as the optimal LNP formulation for potent mRNA delivery in ASCs. Importantly, DIM1T LNPs demonstrated considerably higher mRNA delivery efficiency than MC3, ALC-0315, and SM102, the three state-of-the-art lipid formulations currently approved by the FDA.

To elicit sustained protein biosynthesis within ASCs, we incorporated saRNA in lieu of mRNA into the DIM1T LNP-mediated ASC engineering process. Unlike conventional mRNA, saRNA possesses the inherent capacity for intracellular self-replication through RNA-dependent RNA polymerase, facilitating continuous protein production. However, we observed a substantial impediment in saRNA translation within ASCs with a drastic reduction in translation, which can be attributed to the intracellular host immune defense mechanisms[45,46]. The double-stranded RNA intermediates generated during the replicative translation process activate intracellular RNA sensors, consequently instigating the PKR-eIF2α-mediated translational blockade. To circumvent this repression, we investigated the concomitant delivery of saRNA with an mRNA encoding for an immune evasion protein, VACV protein E3, into ASCs. This approach markedly enhanced and extended the expression of saRNA-encoded luciferase proteins in ASCs, with the expression persisting for 9 days post-treatment. Furthermore, employing DIM1T LNPs for the co-delivery of saRNA/E3 mRNA complexes did not alter the phenotype or the differentiation potential of the engineered ASCs.

Impaired wound healing in diabetic patients is a severe pathological condition where wounds fail to undergo the normal healing process, leading to chronic and highly inflammatory states. Compared with the non-diabetic mice which took around 9 days to achieve complete wound healing, the wounds in the diabetic mice persisted unhealed after 18 days (Fig. 4c; Supplementary Fig. 14a–c).

To explore the therapeutic efficacy of DS-ASCs in diabetic wound healing, we constructed saRNA encoding HGF to generate healing-promoting therapeutic proteins by ASCs. In the murine db/db diabetic wound healing model, the HGF DS-ASCs significantly accelerated the wound healing and promoted the formation of intact epidermis, blood vessels, and myofibroblasts compared with WT ASCs and DIM1T LNP-HGF mRNA treated ASCs. We speculate that the sustained secretion of HGF by the DS-ASCs maintained a relatively high and durable HGF protein level in the wound microenvironment, thus enhancing epithelial repair and neovascularization. Substitution of CXCL12 for HGF in the DIM1T LNPs further improved the wound healing capacity of the engineered ASCs, indicating CXCL12 may be a more suitable therapeutic protein for diabetic wounds. Moreover, compared with CXCL12 DS-ASCs, there were no synergistic therapeutic effects when CXCL12- and HGF-DS-ASCs were combined for treatments. Additionally, when FLuc DS-ASCs were simultaneously administered with CXCL12 DS-ASCs to the wounds, the luminescence intensity within the wounds was consistently higher between Day 3 and Day 9, compared with the combination of FLuc DS-ASCs with WT ASCs (Supplementary Fig. 15). This suggests that the presence of CXCL12 within the wound environment enhances the viability of the transplanted ASCs. Such increased survival may be attributed to the immune-modulatory properties of CXCL12, which remodeled the wound microenvironment, resulting in reduced inflammation and prolonged persistence of ASCs.

The DIM1T LNP-RNA platform can efficiently induce the durable generation of specific proteins in ASCs while minimizing toxicity and phenotypic alterations. Moreover, the LNP-RNA-mediated engineering strategy only takes hours to induce effective protein secretion by ASCs and the secreted protein level can persist 9 days post-treatment. The duration is sufficient for ASCs to accomplish therapeutic efficacy through a 'hit-and-run' functioning mechanism without replacing local tissues, which avoids potential oncogenic effects. Optimization of the current strategy for wound healing promotion may include combinatorial secretion of antibiotic peptides to eliminate bacterial infections at wound sites. Given the simplicity, safety, and feasibility of the LNP-RNA system, our DIM1T LNP-RNA system can be a promising platform to broadly facilitate the generation of various proteins in ASCs based on the therapeutic needs of diverse diseases.

## Methods

### Regulatory

All the mouse studies were approved by the Institutional Animal Care and Use Committee at The Ohio State University (2014A00000106) and The Icahn School of Medicine at Mount Sinai (IPROTO202200000134), and complied with local, state, and federal regulations.

### Chemicals and reagents

D-Lin-MC3-DMA, SM102, ALC-0315 and ALC-0159 were purchased from MedKoo Biosciences. 1,2-Dioleoyl-sn-glycero-3-phosphoethanolamine (DOPE), 1,2-distearoyl-sn-glycero-3-phosphocholine (DSPC), cholesterol, and 1,2-dimyristoyl-rac-glycero-3-methoxypolyethylene glycol-2000 (DMG-PEG$_{2K}$) were purchased from Avanti Polar Lipids. 5-($N$-ethyl-$N$-isopropyl)- Amiloride (EIPA), MβCD, chlorpromazine hydrochloride (CPZ), and 6-(p-Toluidino)-2-naphthalenesulfonic acid

(TNS) were purchased from Sigma-Aldrich. Citrate solutions, Dimethyl sulfoxide, Sodium chloride, sodium phosphate, ammonium citrate, and ammonium acetate were obtained from Thermo Scientific Chemicals. 3-(4,5-Dimethylthiazol-2-yl)-2,5-Diphenyltetrazolium Bromide (MTT) and calcein were purchased from Invitrogen. Compound C16 was obtained from MedChemExpress. Sugar alcohol-derived lipids were purified using column chromatography on a RediSep Gold Resolution silica column (Teledyne Isco) with a CombiFlash Rf system using gradient elution. $^1H$ NMR spectral analyses were conducted using a Bruker Avance 400 MHz device. Mass spectrometry analyses were executed using Acquity SQD UPLC/MS (Waters), LTQ Orbitrap XL mass spectrometer (Thermo Scientific), and the ultrafleXtreme MALDI-TOF mass spectrometer (Bruker) at The Ohio State University.

## Cell culture

293T cells were purchased from ATCC (CRL-3216) and cultured in Dulbecco's modified Eagle medium (DMEM) (Corning) with 10% FBS. Mouse adipose-derived stem cells (ASCs) were obtained by adaptation of previous procedures[62]. Briefly, the inguinal adipose tissues of 8–12 weeks C57BL/6J mice were isolated, minced into small pieces, and digested at 37 °C in phosphate buffer saline (PBS) containing 2% bovine serum albumin (BSA, Sigma-Aldrich) and 2 mg/ml Type I collagenase (Sigma-Aldrich) with constant gentle shaking. DMEM (1 g/L glucose, Corning) containing 10% heat-inactivated fetal bovine serum (Gibco) was added to neutralize the enzyme activity. Then, the cell solution was filtered through a 70 µm nylon filter mesh (BD Falcon) and centrifuged to obtain the stromal vascular fraction (SVF) containing ASCs. The samples were shaken vigorously to ensure the separation of the stromal cells from the primary adipocytes and then centrifuged again to remove the supernatant. The pellets were washed with 10% FBS DMEM medium once and then placed in a single well of 6-well plate per approximately 500 mg of adipose tissue. Cells were subcultured when confluency reached 80–90%. Cells between 3rd to 5th passages were used in the following experiments.

## RNA synthesis

The linear dsDNA of firefly luciferase (FLuc), GFP, VACC E3, mHGF and mCXCL12 were obtained from Integrated DNA Technologies (Supplementary Table 1). The pUC19 vector was used for Gibson assembly to generate the corresponding plasmids. mRNAs and saRNAs were synthesized by previously reported methods[51].

## Formulations of sugar alcohol-derived lipid nanoparticles (LNPs) mRNA

LNPs were formulated via NanoAssemblr (Precision NanoSystems, Canada), by mixing an ethanol solution with ionizable lipids, DOPE, cholesterol and DMG-PEG$_{2k}$, and a mRNA-containing citrate solution (10 mM, pH 3.0). The hydrodynamic diameter, polydispersity index (PDI) and zeta potential were measured by NanoZS Zetasizer (Malvern, USA). Ribogreen assay was used to quantify the encapsulation efficiency. The morphology of DIM1T LNPs was imaged using Glacios Cryo-TEM (Thermo Scientific, USA). The Cryo-EM grids were prepared by adding a small aliquot (3 µl) of sample on a 400-mesh lacey carbon-coated copper grid (Electron Microscopy Sciences). Excess liquid was blotted away. To form a thin film of amorphous ice using Thermo Scientific Vitrobot Mark IV system, the grid was immediately submerged into liquid ethane (CP Lab Safety). Then Cryo-EM images were captured using Falcon 3EC direct electron detector on Glacios Cryo-TEM (linear mode, acceleration voltage of 200 kV) and collected using EPU software (nominal magnification 57,000x).

In the initial screening, newly synthesized ionizable lipids were formulated with DOPE, cholesterol and DMG-PEG$_{2K}$ at the molar ratios of lipid: DOPE: Chol: DMG-PEG$_{2K}$ at 20:30:40:0.75 and FLuc mRNA at the mass ratio of ionizable lipid:mRNA at 10:1. An L16 (4)$^4$ orthogonal table was used to optimize the LNP formulations. Lipofectamine™

3000/mRNA complex was formulated according to the manufacturer's protocol (Thermo Fisher, L3000015). Electroporation for ASCs was performed using P1 Primary Cell 4D-Nucleofector™ X Kit L (Lonza, V4XP-1012). mRNA delivery efficiency was measured after 18 h post-treatment by adding Bright-Glo luciferase substrate (Promega) to cells and quantified using Cytation 5 (Biotek).

The TNS assay was employed to measure the pK$_a$ of LNP formulations[50]. Briefly, mRNA formulations at the concentration of 30 µM total lipids and 6 µM TNS probe were incubated with a series of buffer solutions containing 150 mM sodium chloride, 20 mM sodium phosphate, 25 mM ammonium citrate, and 20 mM ammonium acetate (pH ranging from 2.5 to 11 in increments of 0.5 pH). The mean fluorescence intensity was measured by Cytation-5 (Biotek) at λEx = 321 nm and λEm = 445 nm. The pKa was determined as the pH at which the fluorescence intensity reached half of its maximum value.

## Characterization of DIM1T-FLuc SEC-engineered ASCs

To evaluate the cytotoxicity of DIM1T LNPs in ASCs, the cells were first seeded in a 96-well plate and allowed to culture overnight. Following this, they were treated with either PBS or DIM1T LNPs at various RNA concentrations for 24 h. After the treatment period, MTT reagents were added directly to the cells without changing the medium, and the cells were further incubated at 37 °C for 4 h. Subsequently, the old growth medium was replaced with dimethyl sulfoxide, and the cells were subjected to a 10-min shaking period. The absorbance was then measured at a wavelength of 570 nm to quantify the results, which indicates the level of cytotoxicity of the DIM1T LNPs in the ASCs.

To investigate the specific endocytic pathways utilized by DIM1T LNPs to enter ASCs, various endocytosis inhibitors were used to treat the cells. These inhibitors are EIPA (micropinocytosis inhibitors), MβCD (caveolae-mediated endocytosis inhibitors), and CPZ (clathrin-mediated endocytosis inhibitors). All inhibitors tested resulted in approximately 90% cell viability at the concentration used. The LNPs were added to the ASCs in the presence of various inhibitors after 1 h post-inhibitor treatments. 3 h later, the cells were harvested and analyzed using flow cytometry. In the endosomal escape assay, ASCs were cultured in individual chambers of an imaging dish (Ibidi 80416). After seeding, these cells were treated with calcein and LNPs containing Alexa-Fluor 647-labeled RNA, or solely with calcein, for one hour. Following this incubation, the cells underwent a wash with PBS and were then examined using a Nikon A1R Live Cell Imaging Confocal Microscope.

The cell surface marker profiles and phosphorylation level of PKR and EIF2α of ASCs were profiled using immunophenotyping before and after DIM1T-FLuc SEC LNPs treatments. 24 h after LNPs treatment, cells were harvested, and stained using the following antibodies for flow cytometry analysis. For cell surface markers investigation, CD106(429), CD44 (NIM-R8), SCA-1 (D7), CD29 (HMβ1-1), CD11b (M1/70) and CD45 (S18009D) were purchased from BioLegend Inc. The staining was performed at 1:100 staining for all antibodies. For evaluating the phosphorylation level of PKR and EIF2α in ASCs, PKR antibody (EPR19374, 1: 100 dilution) was purchased from Abcam Inc; Phospho-PKR (Thr446+Thr451, 1: 100 dilution) antibody was purchased from Bioss Inc; EIF2α (CL488-11170, 1: 50 dilution) and phospho-EIF2α (CL488-68023, 1: 50 dilution) were purchased from Proteintech Inc. To analyze PKR and eIF2α expressions in the SEC-ASCs, FLuc saRNA/E3 mRNA was delivered to ASCs using DIM1T LNPs. After 48 h post-treatment, the cells were intracellularly stained for PKR, eIF2α and their phosphorylated counterparts. Using flow cytometry, the ASCs were quantified for PKR and eIF2α expression level. The differentiation ability of DS-ASCs was investigated by a mouse MSC functional identification kit (R&D Systems)[63].

## Expression duration of therapeutic proteins in DS-ASCs

ASCs were seeded in a 12-well plate at 2.0 ×10$^4$ cells/well. After overnight culture, the medium of each well was changed with the fresh

medium containing DIM1T-SEC LNPs or controls. The cells receiving C16 PKR inhibitors were treated with C16 for 1 h before the addition of LNPs. For the SEC group, the mass ratio of saRNA: E3 mRNA is 1:2. Each group was treated with identical total RNA dosage (50 ng/10000 cells). For FLuc saRNA, the cells were trypsinized at designated days for luciferase assay tests. For HGF and CXCL12 saRNA, the conditioned medium was obtained at designated days with the addition of fresh culture medium. The conditioned medium was stored at −80 °C prior to use. The ELISA kits (RayBiotech) were used to determine the concentration of HGF and CXCL12 according to manufacturer protocols, respectively. To measure the intracellular level of E3 proteins, the ASCs receiving DIM1T-SEC LNPs were harvested to prepare cell lysates for ELISA assay. The anti-Vaccinia virus (VACV) E3 Polyclonal antibody (1:2000 dilution, Cusabio CSB-PA322729ZA01VAA) was used to detect E3 proteins based on manufacturer protocols. A Cytation 5 cell imaging multi-mode reader (Biotek) was used to quantify the spectral profile of colorimetric solutions.

### In vivo skin wound healing study in diabetic mice
All male and female db/db mice (BKS.Cg-Dock7m$^{+/+}$ Leprdb/J, 00642, 10–12 weeks) as well as male C57BL/6J mice (6–8 weeks) were purchased from the Jackson Laboratory. Mouse experiments were carried out based on the reported methods[41,54,55]. Mice were housed under a barrier environment (~20 °C, ~45% humidity, and 12/12 light/dark cycle). After wound generation, up to 2 mice were housed in each cage.

To create two full-thickness wounds, db/db mice were anesthetized using isoflurane and had two circular 6 mm biopsy punches (Integra Miltex) applied to their disinfected dorsum. The wound area was affixed with a 7-mm donut-shaped silicone splint (Grace Bio-Labs) and interrupted using 6-0 nylon sutures (Ethicon). HyStem®-HP hydrogels (Advanced BioMatrix) were prepared according to the manufacturer's instructions. The hydrogel solution was used to resuspend ASCs pellet at $3 \times 10^4$ cells/μl within 10 min before treatment. 10 μL of the hydrogel solution containing $3 \times 10^5$ ASCs was then administrated to the wound. To study the synergistic effects, $1.5 \times 10^5$ HGF DS-ASCs were mixed with $1.5 \times 10^5$ CXCL12 DS-ASCs before administration. The wounds were covered with Tegaderm dressing that would be changed every three days. The wounds were imaged until full re-epithelialization. The digital images of wounds were used for wound size measurement based on the inner area of the silicone rings using Adobe Photoshop software. The wound area of each wound was then normalized against the original wound areas. The area under the curve (AUC) of individual wounds in the treatment groups was normalized against the mean AUC of hydrogel vehicle controls within the same experimental runs.

To investigate the persistence of ASCs at wound sites, $1.5 \times 10^5$ FLuc DS-ASCs were mixed with either $1.5 \times 10^5$ CXCL12 DS-ASCs or WT ASCs before being embedded to the wounds on db/db mice. At different time points, D-luciferin substrate solution (Revvity) was intraperitoneally injected into the mice for later luminescence imaging.

### Histology
Wound tissues were harvested with an 8-mm punch biopsy tool (Integra Miltex), fixed using 4% paraformaldehyde (Sigma-Aldrich), and embedded with paraffin (Thermo Scientific Chemicals). Masson's trichrome staining, H&E, and immunofluorescence (IF) were performed on 5μm sections. For IF, the tissue sections were antigen-retrieved with citrate buffer (10 mM, 0.05% Tween 20, pH 6.0) in a microwave with high power for 3 mins and maintained at 95 °C in a steamer for 15 min. The following primary antibodies are used for IF: rabbit mAB to IL-6 (1:50, Abcam ab290735), rabbit pAB to IL-10 (1:200, Abcam ab9969), rabbit mAB CD31 (1:200, Abcam ab222783), and goat pAB αSMA (1:200, Abcam ab7817). Sections were first blocked for 1 h at room temperature (r.t.) and then incubated with

primary antibodies at 4 °C overnight. The secondary antibodies used for 1 h at r.t.: Alexa Fluor 594 donkey anti-rabbit IgG (1:200, Invitrogen, A21207), Alexa Fluor 488 donkey anti-rabbit IgG (1:800, Invitrogen, A21206). 4′,6-diamidino-2-phenylindole (DAPI, Thermo Scientific Chemicals) was used to counterstain nuclear. Digital images of MTS and IF images of CD31, αSMA, IL-6, and IL-10 were quantified using ImageJ.

### Statistics & reproducibility
Statistical analyses are performed on pooled data from at least 2 independent experiments with 4-5 mice per group for animal studies and are all detailed in the corresponding figure legends. The figures display all data points, and no data were excluded from the analysis. The analysis procedures are described in the figure legends.

### Reporting summary
Further information on research design is available in the Nature Portfolio Reporting Summary linked to this article.

## Data availability
All data supporting the findings of this study are available within the article and its supplementary files. Any additional requests for information can be directed to, and will be fulfilled by, the corresponding author. Source data are provided with this paper.

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

## Acknowledgements

Y.X., Y.Z., and Y.Z. contributed equally to this work. The authors acknowledge the use of the core facility provided by the Campus Microscopy & Imaging Facility at Ohio State University. Electron microscopy was performed at the Center for Electron Microscopy and Analysis (CEMAS) at The Ohio State University. Y.D. acknowledges the support from the Maximizing Investigators' Research Award R35GM119679 and R35GM144117 from the National Institute of General Medical Sciences. D.J.I. is an investigator of the Howard Hughes Medical Institute.

## Author contributions

Y.X., Y.Zhang., and Y.Zhong. conceived the work, performed the experiments, analyzed the data and wrote the paper. S.D., X.H., and W.L. contributed to the animal studies. H.L., S.W., C.W., J.Y., and D.D.K. contributed to the mRNA synthesis, LNP characterization and flow cytometry assays. B.D. and D.W.M. contributed to the cryo-TEM imaging. D.J.I and R.W. contributed to the saRNA plasmid. Y.D. conceived and supervised the project and wrote the paper. The final paper was edited and approved by all authors.

## Competing interests

Y.X., Y.Zhang., D.J.I., R.W., and Y.D. are inventors on a patent application (63/433,109) filed by The Ohio State University and Massachusetts Institute of Technology. The patent covers engineered ASCs and their uses in this work. Y.D. is a scientific advisory board member and holds equity in Arbor Biotechnologies and Sirnagen Therapeutics. Y.D. is a co-founder and holds equity in Immunanoengineering Therapeutics. D.J.I. and R.W. are scientific advisory board members and hold equity in Strand Therapeutics. Other authors declare no conflict of interest.
