## [Peer Review File · Nature Communications]

REVIEWER COMMENTS

Reviewer #1 (Remarks to the Author):

This manuscript described the development of a new class of isomannide-derived lipid nanoparticles (DIM1T LNP) and showed its efficiently deliver RNAs to adipose stem cells (ASCs). self-amplifying RNA was chosen as the target since it can produce the protein longer than the traditional mRNA. In order to suppress the immune response triggered by the saRNA, mRNA encoding mRNA encoding an immune-evasive protein E3 was also included in the LNP/mRNA formulation. The authors demonstrated that the delivery of saRNA encoding HGF and CXCL12, together with E3 enzyme into ASCs can extend the protein expression and promote the wound healing in mouse model. The idea to co-express E3 enzyme to mitigate the immune response triggered by saRNA is novel. The data are convincing with necessary controls. I recommend its acceptance after addressing following comments.

1. In figure 2a, please use the raw luminescence intensity rather than the normalized against the Lipofectamine control.
2. the experiments showed that incorporating E3 mRNA extends the protein production (Luciferase and HGF) up to 9 days, while the regular RNA or saRNA alone can only go a day or less. Since the authors used regular mRNA for E3 expression, how long is the E3 half life? If the E3 protein half life is shorter than 9 days, how to explain that the saRNA expression even without the existence of E3 protein.

Reviewer #2 (Remarks to the Author):

Here, Xue et al. investigate a novel LNP formulation of isomannide-derived lipids that efficiently deliver RNA cargo to adipose stem cells in the context of skin wound healing. They observe that a specific formulation (DIM1T LNP) resulted in enhanced ASC transfection efficiency than the state-of-the-art LNP formulations, and that including E3 mRNA in the LNP formulations enhanced the duration of protein expression in ASCs. The also observed that saRNA encoding either HGF or CXCL12 induced superior wound healing over wild-type treatments. This is an exciting application of ex vivo saRNA-transfected ASCs to wound healing, and I believe will be of interest to a broad readership. Overall, the manuscript is well-written and most of the claims are supported by the data. I suggest the following revisions:

Page 3: What do you mean by 'delicate' designs? Please clarify.

Page 3: Are there advantages of this platform over viral vectors? Please include more information in the introduction.

Page 4: Were the formulations made with the molar compositions in the FDA-approved lipid formulations? If not, then please adjust claims about superiority.

Page 4: ... three sets of ionizable lipids (DIM, DIS, and LIS) were prepared with isosorbide,...The author defined the newly-synthesized lipids as "ionizable", is there experimental characterization on their pKa?

Page 5: Please check the term use of “mass spectroscopy”, instead, should it be “mass spectrometry”. Besides, it could be useful for the readership for know which MS technique (ESI) was applied herein.

Page 5: (Figure 1b, step 5) Can the authors comment on the purity of the new (presumably ionizable) lipids? Could the final product be a mixture with various number of aliphatic arms?

Page 5: What do you mean by ‘bio-membrane’? Please clarify.

Page 5: Please include rational for ranges of each lipid included in the orthogonal table.

Page 6: ...The particle size of formulation DIM1T was around... The term us of size is vague here, it would be more informative for the reader if “hydrodynamic diameter” is used herein.

Page 6: Figure 2b- it would be informative to report the N/P ratio instead of wt/wt of DIM/mRNA

Page 7: Suggest including more details about how the analysis of PKR and eIF2a was performed, both in the text and Fig. S5a caption.

Page 9: The experimental details and discussion of inhibitor assay is missing. In Figure 3h, the authors presented the results on inhibitor assay but the rationale of choosing EIPA and CPZ (out of tons of other inhibitors) were not mentioned and the results were not discussed. Besides, experimental details of how such assay were conducted are missing. Please also double check the mechanism of M β CD. Popular view in the field believes that it depletes cholesterol from the cellular membrane and block the lipid raft mechanism of cell uptake. Given the presence of cholesterol in LNPs, could the author comment how M β CD could interact with LNPs? I assume M β CD was first washed away before introducing LNPs. But again, this experimental procedure should appear somewhere.

Page 9: the calcein assay. The authors used the calcein assay to indicate cytosolic release of LNPs since the endosomal escape of LNPs induced the diffused fluorescence pattern of calcein. However, could the author explain why similar diffused fluorescence pattern from the AF647 channel is not so obvious?

Page 10: Figure 3g- The author didn't mention how the cell viability was tested. Usually, people do MTS/MTT assay to generate a sigmoidal curve to quantify the IC50 of the cytotoxicity of substance.

Page 11: What do you mean by ‘the immune regulating capacity of ASCs may reinforce efficacy’? Please clarify.

Page 11: Suggest quantifying the ‘differences in wound healing rates’ and reporting directly in the text.

Page 11: The DS-ASCs yield the most efficient wound-healing kinetics, but there's not much of a different between the mRNA and the saRNA. Could this be a non-specific RNA effect? Suggest including a scrambled or irrelevant RNA group to show that this is not the case.

Page 13: Since both the HGF and CXCL12 enhanced the wound-healing, suggest including a group with both treatments to investigate whether there are synergistic effects.

Page 16: Suggest including more of a discussion on why there were not differences between the mRNA and saRNA groups.

Page 16: Need to include more information on the mRNA and saRNA sequences used.

Page 18: What was the rationale for using only male mice? Given the sex differences between male and female mice in the context of immune responses, it is advisable to use a 50:50 ratio of male to female.

Page 19: Are these the only conflicts of interest?

Figure 2c: Suggest alternative phrasing for 'Molars of DIM1, DOPE, etc.' for example, molar equivalents.

Figure 3e: Suggest labelling each surface marker with the corresponding phenotype.

Figure 4 b/c/d: Is it possible to include a control group of non-db/db for relative wound healing rates? This would be helpful as a baseline for the field.

Reviewer #3 (Remarks to the Author):

Xue et al in "LNP-RNA-engineered adipose stem cells for accelerated diabetic wound healing" describe their work identifying an LNP based process to generate longer expression of mRNAs in adipose stromal cells. They then validate the effectiveness of such RNA engineered cells in a diabetic mouse model of wound healing. Overall the work is logically performed and presented with sufficient experimental replicas. The results in the mouse wound healing models are quite impressive. I have several concerns that the authors should consider addressing:

1. They should in their LNP in vitro experiments replace luciferase with GFP to determine the total number/percentage of cells transduced. Using luciferase one just sees a population average and it would be good to see a population distribution.
2. They should include electroporation controls throughout the experimental strategy. There is no data to support LNP is better than electroporation. There is reason to believe that LNPs may exert a pro-inflammatory aspect. The comparison should be done with identical RNA formulations as the authors have correctly identified a PKR response that can limit effectiveness.
3. In the wound healing model, they should also use a luciferase transfected ASC control to determine persistence of cells at the site of the wound. Especially in combination with their HGF or CXCL12 cells. It is possible that one of the mechanisms of action is that HGF and CXCL12 prolong the duration of the cells in the wound.
4. Minor concern: why did the authors choose to use the term ASC rather than adipose derived MSC?

We thank the editor and reviewers for the helpful feedback. We have included point-by-point responses to address the reviewers' comments in this file. In the revised manuscript, we have added new experimental data and more detailed descriptions, such as E3 protein kinetics, endocytosis assay, synergistic effects of HGF and CXCL12, gender effects on wound healing, and persistence of DS-ASCs in wounds, to support our conclusions. Additionally, we have reorganized the figures to include new results. Revisions in the manuscript suggested by the reviewers have strengthened our manuscript.

REVIEWER COMMENTS

Reviewer #1 (Remarks to the Author):

This manuscript described the development of a new class of isomannide-derived lipid nanoparticles (DIM1T LNP) and showed its efficiently deliver RNAs to adipose stem cells (ASCs). self-amplifying RNA was chosen as the target since it can produce protein longer than the traditional mRNA. In order to suppress the immune response triggered by the saRNA, mRNA encoding mRNA encoding an immune-evasive protein E3 was also included in the LNP/mRNA formulation. The authors demonstrated that the delivery of saRNA encoding HGF and CXCL12, together with E3 enzyme into ASCs can extend the protein expression and promote the wound healing in mouse model. The idea to co-express E3 enzyme to mitigate the immune response triggered by saRNA is novel. The data are convincing with necessary controls. I recommend its acceptance after addressing following comments.

We thank the reviewer for the encouraging comments.

1. In figure 2a, please use the raw luminescence intensity rather than the normalized against the Lipofectamine control.

We have changed Figure 2a with the raw luminescence intensity.

Figure 2. a, Fluc-mRNA delivery efficiency in primary murine ASCs represented as luminescence intensity. Data are from $n = 3$ biologically independent samples and are presented as mean \pm s.d. Statistical significance is analyzed by one-way ANOVA followed by Dunnett's multiple comparison test. *** $P < 0.001$, **** $P < 0.0001$.

2. the experiments showed that incorporating E3 mRNA extends the protein production (luciferase and HGF) up to 9 days, while the regular RNA or saRNA alone can only go a day or less. Since the authors used regular mRNA for E3 expression, how long is the E3 half life? If the E3 protein half life is shorter than 9 days, how to explain that the saRNA expression even without the existence of E3 protein.

We have performed an additional experiment to quantify the intracellular levels of E3 proteins in ASCs treated with DIM1T-FLuc SEC. As shown in **Supplementary Fig. 7a**, intracellular expression of E3 proteins was detectable even 9 days post-treatment. This expression level may facilitate the expression of saRNA.

Supplementary Fig. 7: a, Expression kinetics of E3 protein delivered by DIM1T LNPs in ASCs. The cell lysate was collected on Days 1, 2, 3, 5, 7, and 9. The E3 protein level in the cell lysates was analyzed using an ELISA assay. All data are presented as mean \pm s.d.

Reviewer #2 (Remarks to the Author):

Here, Xue et al. investigate a novel LNP formulation of isomannide-derived lipids that efficiently deliver RNA cargo to adipose stem cells in the context of skin wound healing. They observe that a specific formulation (DIM1T LNP) resulted in enhanced ASC transfection efficiency than the state-of-the-art LNP formulations, and that including E3 mRNA in the LNP formulations enhanced the duration of protein expression in ASCs. They also observed that saRNA encoding either HGF or CXCL12 induced superior wound healing over wild-type treatments. This is an exciting application of ex vivo saRNA-transfected ASCs to wound healing, and I believe will be of interest to a broad readership. Overall, the manuscript is well-written and most of the claims are supported by the data. I suggest the following revisions:

We thank the reviewer for the constructive feedback.

Page 3: What do you mean by 'delicate' designs? Please clarify.

We have included additional descriptions to describe our design.

'Specifically, fine-tuning the structures of ionizable lipids, such as head groups and hydrophobic tails, critically influences the RNA delivery efficiency of the formulated LNPs and their affinity for specific cell types.'

Page 3: Are there advantages of this platform over viral vectors? Please include more information in the introduction.

We have included the following information in the introduction:

Although viral vectors can induce long-term expression of functional proteins in stem cells including ASCs, their persistence in the cells may elevate the risk of insertional mutagenesis in recipients after administration, potentially leading to malignant transformation. Unlike viral vectors, LNP carriers exhibit great biocompatibility and biodegradability. Moreover, their delivery efficiency can be readily optimized by adjusting the molar ratios of individual formulation components. Additionally, with the advancements in formulation methodologies, LNP can be manufactured in a large scale with high batch-to-batch reproducibility.

Page 4: Were the formulations made with the molar compositions in the FDA-approved lipid formulations? If not, then please adjust claims about superiority.

The formulation compositions of DLin-MC3-DMA, ALC-0315, or SM102, the FDA-approved ionizable lipids, were formulated with the FDA-approved formulation compositions and ratios. The LNPs formulated based on sugar alcohol-derived ionizable lipids in this study are composed of DOPE, Cholesterol and DMG-PEG_{2k} with an optimal formulation ratio identified in the experiments. Detailed formulation information has been included in the method section.

Page 4: ... three sets of ionizable lipids (DIM, DIS, and LIS) were prepared with isosorbide,... The author defined the newly-synthesized lipids as "ionizable", is there experimental characterization on their pKa?

We have used the TNS assay to measure the pKa of DIM1T LNPs, which is 6.56. Experimental details of the TNS assay were included in the method section.

Supplementary Fig. 3: c, Representative TNS assay curves for determining the apparent pK_a of DIM1T LNPs.

Page 5: Please check the term use of “mass spectroscopy”, instead, should it be “mass spectrometry”. Besides, it could be useful for the readership for know which MS technique (ESI) was applied herein. We have changed “mass spectroscopy” to “mass spectrometry” and included the use of electrospray ionization (ESI) mass spectrometry in the method section.

‘Mass spectrometry analyses were executed using Acquity SQD UPLC/MS (Waters), LTQ Orbitrap XL mass spectrometer (Thermo Scientific), and the ultrafleXtreme MALDI-TOF mass spectrometer (Bruker) at The Ohio State University.’

Page 5: (Figure 1b, step 5) Can the authors comment on the purity of the new (presumably ionizable) lipids? Could the final product be a mixture with various number of aliphatic arms? Based on analysis of the nuclear magnetic resonance (NMR) data, the purity of these lipids is estimated to be $\geq 95\%$.

Page 5: What do you mean by ‘bio-membrane’? Please clarify. Bio-membrane refers to cell membrane or endosome membrane. We have included an additional description.

‘As the hydrophobic tails on ionizable lipids can greatly affect LNP formulations and their interaction with bio-membranes such as cell membranes and endosome membranes, we installed five types of hydrophobic tails to the core amine 4 to equip DIM lipids via epoxide ring-opening reaction with alkyl epoxide or reductive amination reaction with aldehydes.’

Page 5: Please include rational for ranges of each lipid included in the orthogonal table. For ionizable lipids, their molar ratio in the formulation as well as the N/P ratio to mRNA has been taken into consideration based on the numbers of amino groups in the chemical structure. Our previous study reported that TT3, a lipid-like material that contains 3 amino groups, displayed the highest delivery efficiency at the molar ratio of 20 in the formulation. Since DIM1 contains 2 amino groups, we speculate that molar ratios ranging from 10 to 40 may cover the optimal one.

For helper lipids (DOPE and Chol) and PEG-lipids, we refer to their molar ratios used in clinical-approved LNPs as well as our previous studies to determine the ranges.

Page 6: ...The particle size of formulation DIM1T was around... The term us of size is vague here, it would be more informative for the reader if “hydrodynamic diameter” is used herein.

We have clarified the description of hydrodynamic diameter accordingly.

Page 6: Figure 2b- it would be informative to report the N/P ratio instead of wt/wt of DIM/mRNA

We have included the N/P ratio in the revised manuscript.

Optimization table of DIM1 LNPs							
Round	Levels	Factors (molar ratio)				DIM1/mRNA (wt/wt)	DIM1/mRNA (NP ratio)
		DIM1	DOPE	Chol	PEG		
1st	1	10	20	30	0.25	10	6.7
	2	20	30	40	0.50	10	6.7
	3	30	40	50	0.75	10	6.7
	4	40	50	60	1.00	10	6.7
2nd	DIM-1Q	30	20	40	0.75	5	3.3
	DIM-1R	30	20	40	0.75	7.5	5.0
	DIM-1S	30	20	40	0.75	10	6.7
	DIM-1T	30	20	40	0.75	12.5	8.4
	DIM-1U	30	20	40	0.75	15	10.0
	DIM-1V	30	20	40	0.75	17.5	11.7
	DIM-1W	30	20	40	0.75	20	13.4

Figure 2: b, Table for the two rounds of DIM1/FLuc-mRNA LNP optimization. Chol, cholesterol; PEG, DMG-PEG_{2k}.

Page 7: Suggest including more details about how the analysis of PKR and eIF2 α was performed, both in the text and Fig. S5a caption.

We have added the following descriptions in the revised manuscript.

‘To analyze PKR and eIF2 α expressions in the SEC-ASCs, FLuc saRNA/E3 mRNA was delivered to ASCs using DIM1T LNPs. After 48h post-treatment, the cells were intracellularly stained for PKR, eIF2 α and their phosphorylated counterparts. Using flow cytometry, the ASCs were quantified for PKR and eIF2 α expression level.’

Page 9: The experimental details and discussion of inhibitor assay is missing. In Figure 3h, the authors presented the results on inhibitor assay but the rationale of choosing EIPA and CPZ (out of tons of other inhibitors) were not mentioned and the results were not discussed. Besides, experimental details of how such assay were conducted are missing. Please also double check the mechanism of M β CD. Popular view in the field believes that it depletes cholesterol from the cellular membrane and block the lipid raft mechanism of cell uptake. Given the presence of cholesterol in LNPs, could the author comment how M β CD could interact with LNPs? I assume M β CD was first washed away before introducing LNPs. But again, this experimental procedure should appear somewhere.

We have described the results and added the experimental details of the inhibitor assay in the revised manuscript.

‘To investigate the specific endocytic pathways utilized by DIM1T LNPs to enter ASCs, various endocytosis inhibitors were used to treat the cells. These inhibitors are EIPA (micropinocytosis inhibitors), M β CD (caveolae-mediated endocytosis inhibitors), and CPZ (clathrin-mediated endocytosis inhibitors). All inhibitors tested resulted in approximately 90% cell viability at the concentration used.’

The LNPs were added to the ASCs in the presence of various inhibitors after 1h post-inhibitor treatments. 3 hours later, the cells were harvested and analyzed using flow cytometry.'

The choice of CPZ, EIPA, and M β CD was based on their specificity to their respective pathways. They are also widely used for endocytosis investigations in the fields (PMID: 20226220; PMID: 31907443; PMID: 28919558; PMID: 31239678).

We chose not to wash away the inhibitors before the LNP treatment because the cells might regain their uptake ability without inhibitors. To investigate whether M β CD affects the LNP structure, we used another caveolin inhibitor, genistein, for comparison. Unlike M β CD, genistein doesn't disrupt cholesterol in the cell membrane; instead, it hinders Src kinase-dependent phosphorylation of caveolin-1 and inhibits vesicle fusion, potentially avoiding adverse effects on LNPs (PMID: 20226220; PMID: 22973022). Under the same experimental conditions, we observed a comparable decrease in the cellular uptake efficiency of DIM1T LNPs in ASCs treated with either M β CD or genistein. Moreover, altering the medium prior to introducing the LNPs partially restored the ASCs' uptake capacity.

Extended Data Fig. 1: Uptake efficiency of DIM1T LNPs loaded with Alexa-Fluor 647 labeled RNAs in ASCs treated by M β CD or genistein. M β CD (w) and Genistein (w) mean washing away medium containing inhibitors before the addition of DIM1T LNPs.

Page 9: the calcein assay. The authors used the calcein assay to indicate cytosolic release of LNPs since the endosomal escape of LNPs induced the diffused fluorescence pattern of calcein. However, could the author explain why similar diffused fluorescence pattern from the AF647 channel is not so obvious?

We have reproduced the experiments and provided a clear image in the revised manuscript.

Figure 3: i, CLSM images of ASCs stained with calcein alone or with DIM1T-LNPs (scale bar = 50 μ m).

Page 10: Figure 3g- The author didn't mention how the cell viability was tested. Usually, people do MTS/MTT assay to generate a sigmoidal curve to quantify the IC50 of the cytotoxicity of substance. We have added the experimental details of the cell viability assay in the method section.

'To evaluate the cytotoxicity of DIM1T LNPs in ASCs, the cells were first seeded in a 96-well plate and allowed to culture overnight. Following this, they were treated with either PBS or DIM1T LNPs at various RNA concentrations for 24 hours. After the treatment period, MTT reagents were added directly to the cells without changing the medium, and the cells were further incubated at 37°C for 4 hours. Subsequently, the old growth medium was replaced with dimethyl sulfoxide, and the cells were subjected to a 10-minute shaking period. The absorbance was then measured at a wavelength of 570 nm to quantify the results, which indicates the level of cytotoxicity of the DIM1T LNPs in the ASCs.'

Page 11: What do you mean by 'the immune regulating capacity of ASCs may reinforce efficacy'? Please clarify.

To avoid confusion, we have removed this statement.

Page 11: Suggest quantifying the 'differences in wound healing rates' and reporting directly in the text. We have quantified the healing rates and described them in the revised manuscript.

Page 11: The DS-ASCs yield the most efficient wound-healing kinetics, but there's not much of a difference between the mRNA and the saRNA. Could this be a non-specific RNA effect? Suggest including a scrambled or irrelevant RNA group to show that this is not the case.

We have reproduced the wound healing model that includes FLuc DS-ASCs as a treatment group. As shown in **Supplementary Fig. 10a-b**, the wounds treated with CXCL12 DS-ASCs or CXCL12/HGF DS-ASCs groups demonstrated significantly accelerated healing rate and smaller wound healing AUC compared with those receiving FLuc DS-ASCs treatment. Particularly, the average relative wound size of these groups on Day 15 was 28.87%±0.066% (Vehicles), 20.00%±0.072% (FLuc DS-ASCs), 0.00%±0.000% (CXCL12 DS-ASCs) and 0.013%±0.019% (CXCL12/HGF DS-ASCs). Moreover, a full 100% of the wounds in CXCL12 DS-ASCs group and 80% in CXCL12/HGF SC-ASCs group exhibited complete closure by Day 15 (**Supplementary Fig. 10c**). However, none of the wounds in mice treated with either vehicles or FLuc DS-ASCs achieved complete healing. Therefore, the healing efficacy in diabetic wounds results from the sustainable generation of therapeutic proteins by the DIM1T-SEC engineering ASCs.

Supplementary Fig. 10: **a**, Relative wound size on the mice receiving vehicle controls, Fluc DS-ASCs, CXCL12 DS-ASCs, and CXCL12/HGF DS-ASCs. **b**, Mean AUC of individual wounds of each DS-ASC group normalized to vehicle controls. **c**, The complete wound closure time in vehicle controls, Fluc DS-

ASCs, CXCL12 DS-ASCs, and CXCL12/HGF DS-ASCs. The significant differences in time to closure between groups are analyzed using the Log-rank test. Statistical significance and P values in **a-c** are analyzed by one-way ANOVA followed by Dunnett's multiple comparison test. n.s. not significant, $P > 0.05$, $**P < 0.01$, $****P < 0.0001$.

Page 13: Since both the HGF and CXCL12 enhanced the wound-healing, suggest including a group with both treatments to investigate whether there are synergistic effects.

To investigate the potential synergistic effects of CXCL12 and HGF in wound healing, we have conducted the same excisional wound healing experiments in db/db mice and embedded the engineered DS-ASCs generating both CXCL12 and HGF on the wound sites. In comparison with vehicle controls and FLuc DS-ASCs group, the wounds treated with the CXCL12/HGF DS-ASCs or CXCL12 DS-ASCs demonstrated pronounced acceleration in the healing process, which was highlighted by the decrease in wound size and smaller wound healing AUC (**Supplementary Fig. 9 and 10a-b**). Remarkably, 80% of the wounds in CXCL12/HGF SC-ASCs group and a full 100% in CXCL12 DS-ASCs group exhibited complete closure by Day 15 (**Supplementary Fig. 10c**). Moreover, both CXCL12 DS-ASCs and CXCL12/HGF DS-ASCs substantially increased the thickness of the hyperproliferative epidermis and density of CD31⁺ blood vessels and α SMA⁺ myofibroblast relative to vehicles and FLuc-DS ASCs (**Supplementary Fig. 10d-i**). Both groups also showed increased accumulation of IL10 as well as a decreased level of IL6 in the wound microenvironment (**Supplementary Fig. 10j-m**). However, no statistically significant differences were observed in the aforementioned parameters between these two groups, suggesting the absence of synergistic therapeutic efficacy when combining CXCL12 and HGF in expediting wound healing.

Supplementary Fig. 9: Representative digital images of the wounds of each group from the study on synergistic effects of CXCL12 and HGF in wound healing.

Supplementary Fig. 10: Studies on synergistic effects of CXCL12 and HGF in wound healing. **a**, Relative wound size on the mice receiving vehicle controls, FLuc DS-ASCs, CXCL12 DS-ASCs, and CXCL12/HGF DS-ASCs. **b**, Mean AUC of individual wounds of each DS-ASC group normalized to vehicle controls. **c**, The complete wound closure time in vehicle controls, FLuc DS-ASCs, CXCL12 DS-ASCs, and CXCL12/HGF DS-ASCs. The significant differences in time to closure between groups are analyzed using the Log-rank test. ** $P < 0.01$, **** $P < 0.0001$. **d**, Representative MTS and H&E images of wounds on D15 for each group. **e**, Quantification of the thickness of epidermis on the wound tissues from each group. **f**, **h**, **j**, and **l**, Representative CD31⁺, αSMA⁺, IL-6 and IL-10 IF images of wounds on D15 for each group. **g**, **i**, **k**, and **m**, Quantification of the CD31⁺ cells, the αSMA⁺ cells, the IL-6, and the IL-10. All data are presented as mean ± s.d. Statistical significance and P values are analyzed by one-way ANOVA followed by Dunnett's multiple comparison test. n.s. not significant, $P > 0.05$, ** $P < 0.01$, *** $P < 0.001$, **** $P < 0.0001$. Scale bars, 1 mm (**d**); 50 μm (**f**, **h**, **j**, and **l**).

Page 16: Suggest including more of a discussion on why there were not differences between the mRNA and saRNA groups.

We did observe significant differences between the mRNA (DM-ASCs) and SEC (saRNA/E3 mRNA complex; DS-ASCs) groups (Fig. 4c-k). The HGF DS-ASCs yielded the most efficient wound-healing kinetics as the average wound size of this group on Day 18 was $0.00 \pm 0.00\%$ (Fig. 4c-d). Such rates in other groups were $25.60\% \pm 6.28\%$ (Vehicles), $18.87\% \pm 4.70\%$ (WT ASCs) and $7.03\% \pm 6.29\%$ (DM-ASCs). On Day 18, DS-ASCs group was able to achieve complete wound closure while 60% of the wounds treated by DM-ASCs remained unhealed (Fig. 4e). Moreover, compared with other groups including DM-ASCs, DS-ASCs demonstrated more efficient re-epithelialization and displayed well-formed hyperproliferative epidermis in the wounds as well as higher density of CD31⁺ vessels and α SMA⁺ myofibroblasts (Fig. 4f-k). These data suggested that DS-ASCs, with the capability to sustainably generate therapeutic proteins, are a more efficient treatment platform than DM-ASCs in treating diabetic wounds.

Page 17: Need to include more information on the mRNA and saRNA sequences used.

We have included the coding sequences of each functional protein in mRNA and saRNA sequences.

Supplementary table 1: Coding sequences of mHGF, mCXCL12, and Vaccinia virus (VACV) E3 proteins.

Protein	Coding sequences
mHGF	AUG AUG UGG GGC ACA AAA UUG CUU CCA GUA CUC UUG UUG CAG CAU GUC CUU CUC CAU UU G CUU UUG CUU CAU GUA GCU AUU CCU UAU GCU GAA GGU CAA AAG AAG CGA CGC AAU ACU C UU CAC GAA UUU AAG AAG AGC GCA AAA ACC ACU UUG ACU AAA GAA GAU CCU CUG CUC AAG A UA AAG ACA AAG AAG GUG AAC AGC GCC GAU GAA UGU GCC AAU CGA UGC AUA CGC AAU AGG GGU UUC ACU UUC ACU UGU AAA GCU UUU GUU UUU GAU AAG UCU CGA AAA CGC UGU UAU UG G UAU CCU UUU AAU UCC AUG UCU UCC GGG GUU AAA AAA GGA UUC GGA CAU GAG UUU GAC CUC UAC GAA AAC AAA GAC UAU AUU CGG AAU UGC AUA AUC GGC AAA GGA GGU AGU UAU AAA GGU ACA GUC AGU AUC ACU AAG UCC GGU AUC AAG UGC CAG CCU UGG AAU AGC AUG AUC CC A CAC GAG CAU AGU UUU UUG CCC UCU UCU UAU CGC GGU AAA GAU UUG CAG GAG AAC UAC UGU CGA AAU CCA CGG GGA GAA GAA GGC GGU CCU UGG UGC UUU ACU UCU AAU CCC GAA G UG CGG UAC GAA GUA UGC GAC AUU CCA CAG UGU AGC GAA GUA GAA UGC AUG ACU UGU AAC GGC GAA UCA UAC CGG GGA CCC AUG GAC CAU ACA GAG UCU GGA AAG ACU UGC CAA CGC UG G GAC CAA CAA ACA CCC CAU CGC CAU AAG UUC CUU CCA GAA AGA UAU CCU GAU AAA GGA UU C GAU GAU AAC UAC UGU CGG AAU CCU GAU GGU AAG CCU CGA CCU UGG UGC UAC ACA CUU GAU CCC GAU ACC CCA UGG GAA UAC UGU GCA AUA AAG ACC UGU GCC CAU UCC GCC GUC AA U GAA ACC GAC GUU CCC AUG GAA ACU ACU GAA UGC AUA CAG GGC CAG GGU GAA GGG UAU C GC GGC ACA AGC AAC ACU AUA UGG AAC GGG AUC CCU UGC CAA CGC UGG GAU UCC CAG UAU CCA CAU AAA CAC GAU AUA ACU CCU GAG AAC UUC AAG UGU AAA GAU CUC CGC GAG AAU UAU UGU AGG AAU CCU GAU GGG GCA GAG UCC CCU UGG UGU UUU ACU ACU GAC CCC AAU AUC C GC GUA GGG UAU UGC UCC CAG AUU CCC AAG UGC GAC GUU UCC UCC GGC CAA GAC UGU UA U AGG GGU AAU GGU AAG AAC UAU AUG GGU AAU UUG UCA AAA ACU AGA UCA GGC UUG ACA U GU UCU AUG UGG GAC AAA AAU AUG GAG GAC UUG CAC CGA CAU AUC UUU UGG GAG CCA GAC GCA UCU AAA UUG AAC AAG AAC UAU UGU AGA AAC CCU GAU GAU GAU GCC CAC GGC CCA UG G UGC UAC ACA GGA AAC CCA CUU AUC CCC UGG GAC UAU UGC CCU AUA UCU CGA UGU GAG GGU GAC ACC ACA CCU ACU AUA GUG AAC CUG GAU CAU CCA GUC AUA UCA UGU GCA AAG ACA AAA CAG CUG AGA GUU GUA AAU GGU AUU CCA ACC CAG ACA ACC GUA GGA UGG AUG GUG AG U CUC AAA UAU AGA AAU AAG CAC AUU UGC GGA GGC UCU CUU AUC AAA GAG UCC UGG GUA C UC ACC GCA CGC CAG UGU UUU CCU GCA CGG AAC AAA GAU CUU AAG GAU UAU GAA GCC UGG CUG GGG AUA CAC GAC GUA CAU GAA AGG GGA GAA GAG AAG CGA AAA CAA AUU CUU AAC AUC AGC CAG UUG GUG UAC GGG CCU GAA GGA UCC GAU UUG GUG UUG UUG AAA CUG GCU CGG CCC GCC AUC UUG GAC AAU UUC GUC AGC ACU AUA GAC CUU CCC UCA UAC GGU UGU ACU AU U CCC GAA AAA ACC ACC UGU UCA AUC UAU GGC UGG GGA UAC ACA GGG CUG AUC AAU GCU G AC GGU CUC CUG CGA GUU GCA CAU CUU UAC AUC AUG GGU AAC GAG AAG UGU UCC CAA CAU CAC CAA GGU AAA GUA ACU CUC AAC GAA UCA GAA CUC UGC GCU GGG GCU GAG AAA AUC GG G UCC GGC CCU UGU GAA GGA GAC UAC GGC GGA CCA CUU AUC UGC GAG CAG CAC AAA AUG AGA AUG GUU CUG GGU GUC AUA GUU CCC GGG CGC GGU UGC GCC AUU CCU AAC CGA CCA G GU AUU UUU GUC AGG GUG GCC UAU UAC GCC AAG UGG AUU CAC AAG GUA AUU UUG ACU UAC AAA CUU
mCXCL12	AUG GAU GCC AAA GUG GUA GCC GUU UUG GCA CUU GUA UUG GCC GCA CUC UGU AUU UCA G AU GGG AAA CCC GUA AGC CUG AGU UAC AGG UGC CCC UGC CGG UUU UUU GAG UCA CAU AU C GCC CGA GCU AAU GUG AAG CAU UUG AAG AUC CUU AAC ACU CCU AAC UGC UGC CUG CAG A UA GUG GCC AGA CUG AAG AAC AAU AAC CGA CAG GUU UGC AUC GAC CCA AAA CUG AAG UGG AUA CAA GAG UAU CUU GAG AAG GCU CUU AAC AAA AGA CUU AAA AUG
VACV E3	AUG UCU AAA AUC UAU AUC GAU GAG CGA AGC GAU GCC GAA AUA GUU UGU GCA GCC AUA AA G AAC AUU GGC AUU GAA GGG GCC ACU GCC GCU CAG CUC ACA CGC CAG UUG AAC AUG GAG AAA CGC GAA GUA AAC AAG GCA CUU UAU GAC CUG CAG CGA AGU GCA AUG GUU UAC AGC UC U GAC GAU AUA CCU CCC CGC UGG UUU AUG ACA ACC GAA GCU GAU AAG CCC GAU GCU GAU GCU AUG GCA GAC GUU AUC AUC GAU GAU GUU UCA CGC GAG AAG UCA AUG CGA GAA GAC C AC AAG AGU UUC GAU GAC GUG AUU CCC GCA AAG AAA AUC AUC GAC UGG AAG GAU GCC AAC CCU GUG ACC AUA AUA AAC GAA UAU UGC CAA AUC ACA AAA CGG GAC UGG UCU UUU AGA AU A GAA UCU GUU GGG CCU AGC AAU AGU CCA ACC UUU UAC GCU UGU GUA GAC AUC GAC GGC AGG GUU UUC GAC AAG GCA GAC GGC AAA UCA AAA AGA GAU GCU AAG AAC AAU GCU GCU AA A CUU GCU GUG GAU AAG CUG CUC GGU UAU GUC AUC AUU CGG UUU

Page 18: What was the rationale for using only male mice? Given the sex differences between male and female mice in the context of immune responses, it is advisable to use a 50:50 ratio of male to female.

We have performed a new experiment to study sex differences with the following groups:
(1) Vehicles; (2) FLuc SEC-ASCs; (3) CXCL12 SEC-ASCs; (4) HGF/CXCL12 SEC-ASCs.
Each group contains 2 male mice (4 wounds) and 2 female mice (4 wounds).

As is shown in **Supplementary Fig. 11**, there was no statistical significance in wound size reduction rates between female and male mice in all groups.

Supplementary Fig. 11: Gender effects on wound healing. **a-d**, Relative wound sizes measured in both male and female mice receiving the following treatments: Vehicle, FLuc DS-ASCs, CXCL12 DS-ASCs, and CXCL12/HGF DS-ASCs, respectively. $n = 4$ wounds for each gender in every group. All data are presented as mean \pm s.d. Statistical significance and P values are analyzed by one-way ANOVA followed by Dunnett's multiple comparison test. n.s. not significant, $P > 0.05$.

Page 19: Are these the only conflicts of interest?

We have updated current conflicts of interest.

Figure 2c: Suggest alternative phrasing for 'Molars of DIM1, DOPE, etc.' for example, molar equivalents.
We have corrected the 'molars' to 'molar ratios' in the revised manuscript.

Figure 3e: Suggest labelling each surface marker with the corresponding phenotype.

We have labeled each surface marker with the corresponding phenotype in the revised **Figure 3e**.

CD106: vascular cell adhesion molecule 1 (VCAM-1)

CD29: Integrin beta 1 (ITGB1)

Sca-1: Stem Cell Antigen-1

CD11: Integrin alpha M (ITGAM)

Figure 4 b/c/d: Is it possible to include a control group of non-db/db for relative wound healing rates?
This would be helpful as a baseline for the field.

We have added a non-db/db (WT C57BL6/J) mice group as a baseline for the wound healing rates. As depicted in **Fig. 4c** and **Supplementary Fig. 12a-c**, wound healing kinetics in untreated non-diabetic

mice progressed at a significantly faster rate compared with diabetic mice subjected to various treatments. Remarkably, complete wound closure was observed on Day 9 in 5 out of the 8 wounds evaluated.

Supplementary Fig. 12: Wound healing kinetics of non-diabetic mice (WT C57BL6/J). **a**, Relative size of the wounds on non-diabetic mice without treatments. $n=8$ wounds. **b**, Relative size of each wound. **c**, Representative digital images of the wounds. All data are presented as mean \pm s.d.

Reviewer #3 (Remarks to the Author):

Xue et al in "LNP-RNA-engineered adipose stem cells for accelerated diabetic wound healing" describe their work identifying an LNP based process to generate longer expression of mRNAs in adipose stromal cells. They then validate the effectiveness of such RNA engineered cells in a diabetic mouse model of wound healing. Overall the work is logically performed and presented with sufficient experimental replicas. The results in the mouse wound healing models are quite impressive. I have several concerns that the authors should consider addressing:

We thank the reviewer for the in-depth review.

1. They should in their LNP in vitro experiments replace luciferase with GFP to determine the total number/percentage of cells transduced. Using luciferase one just sees a population average and it would be good to see a population distribution.

We have used DIM1T LNPs to deliver eGFP-mRNA to ASC and quantified the population distribution using flow cytometry. The results have been included in the revised manuscript and supporting information.

'In addition, DIM1T LNPs encapsulating GFP mRNA yielded over 90% GFP positive ASCs, with the mean fluorescence intensity (MFI) notably superior to those FDA-approved lipid formulations (Supplementary Fig. 4a-c).'

Supplementary Fig. 4: DIM1T LNPs deliver GFP mRNA in ASCs. **a**, GFP expression in ASCs treated with DIM1T LNPs encapsulating GFP mRNA after 18h. **b**, Quantification of GFP expression from **a**. **c**, The mean fluorescence intensity (MFI) of GFP in GFP⁺ ASCs. All data are presented as mean \pm s.d. Statistical significance and P values are analyzed by one-way ANOVA followed by Dunnett's multiple comparison test. ****P < 0.0001.

2. They should include electroporation controls throughout the experimental strategy. There is no data to support LNP is better than electroporation. There is reason to believe that LNPs may exert a pro-inflammatory aspect. The comparison should be done with identical RNA formulations as the authors have correctly identified a PKR response that can limit effectiveness.

We have included electroporation data in Figure 2a. The initial screening in Figure 2a was based on FLuc-mRNA delivery efficiency, and DIM1 LNPs showed 75-fold higher intensity than electroporation.

Figure 2. a, Fluc-mRNA delivery efficiency in primary murine ASCs represented as luminescence intensity. Data are from $n = 3$ biologically independent samples and are presented as mean \pm s.d. Statistical significance is analyzed by one-way ANOVA followed by Dunnett's multiple comparison test. *** $P < 0.001$, **** $P < 0.0001$.

3. In the wound healing model, they should also use a luciferase transfected ASC control to determine persistence of cells at the site of the wound. Especially in combination with their HGF or CXCL12 cells. It is possible that one of the mechanisms of action is that HGF and CXCL12 prolong the duration of the cells in the wound.

To evaluate the impact of CXCL12 on the persistence of DS-ASCs, we co-delivered DS-ASCs with either CXCL12 DS-ASCs or WT ASCs. As is shown in **Supplementary Fig. 13**, when FLuc DS-ASCs were combined with CXCL12 DS-ASCs, the luminescence intensity within the wounds was consistently higher between Day 3 and Day 9, compared with the combination of FLuc DS-ASCs with WT ASCs. This suggests that the presence of CXCL12 within the wound environment enhances the viability of the transplanted ASCs. Such increased survival may be attributed to the immune-modulatory properties of CXCL12, which remodeled the wound microenvironment, resulting in reduced inflammation and prolonged persistence of ASCs.

Supplementary Fig. 13: Persistence of DS-ASCs in diabetic wounds. a, Luminescence intensity of the embedded FLuc DS-ASC in wounds from Day 0 to Day 12. **b**, Zoom-in plot of luminescence intensity from Day 6 to Day 12 in **a**. All data are presented as mean \pm s.d. Statistical significance and P values are analyzed by one-way ANOVA followed by Dunnett's multiple comparison test. n.s. not significant, $P > 0.05$, * $P < 0.05$, *** $P < 0.001$.

4. Minor concern: why did the authors choose to use the term ASC rather than adipose derived MSC? The International Fat Applied Technology Society reached a consensus to adopt the term “adipose-derived stem cells” (ASCs) to identify the isolated, plastic-adherent, multipotent cell population from adipose tissues (PMID: 25126376; PMID: 17495232). Therefore, we used ASCs in our study rather than adipose-derived MSCs (AD-MSCs).

REVIEWERS' COMMENTS

Reviewer #1 (Remarks to the Author):

The responses to the reviewers are satisfactory. I recommend its acceptance for publication.

Reviewer #2 (Remarks to the Author):

My concerns have been sufficiently addressed and I recommend publication at this time.